# Proto Successor Measure: Representing the Behavior Space of an RL Agent

**Siddhant Agarwal** [* 1]  **Harshit Sikchi** [* 1]  **Peter Stone** [1 2 †]  **Amy Zhang** [1 †]

## Abstract

Having explored an environment, intelligent agents should be able to transfer their knowledge to most downstream tasks within that environment without additional interactions. Referred to as "zero-shot learning", this ability remains elusive for general-purpose reinforcement learning algorithms. While recent works have attempted to produce zero-shot RL agents, they make assumptions about the nature of the tasks or the structure of the MDP. We present *Proto Successor Measure*: the basis set for all possible behaviors of a Reinforcement Learning Agent in a dynamical system. We prove that any possible behavior (represented using visitation distributions) can be represented using an affine combination of these policy-independent basis functions. Given a reward function at test time, we simply need to find the right set of linear weights to combine these bases corresponding to the optimal policy. We derive a practical algorithm to learn these basis functions using reward-free interaction data from the environment and show that our approach can produce the near-optimal policy at test time for any given reward function without additional environmental interactions. Project page: agarwal-siddhant10.github.io/projects/psm.html

## 1. Introduction

A wide variety of tasks can be defined within an environment (or any dynamical system). For instance, in navigation environments, tasks can be defined to reach a goal, follow a path, reach a goal while avoiding certain states etc. Once familiar with an environment, humans have the wonderful ability to perform new tasks in that environment without any additional practice. For example, consider the last time you moved to a new city. At first, you may have needed

to explore various routes to figure out the most efficient way to get to the nearest supermarket or place of work. But eventually, you could probably travel to new places efficiently the very first time you needed to get there. Like humans, intelligent agents should be able to infer the necessary information about the environment during exploration and use this experience for solving any downstream task efficiently. Reinforcement Learning (RL) algorithms have seen great success at finding a sequence of decisions that optimally solves a given task in the environment (Wurman et al., 2022; Fawzi et al., 2022). In RL settings, tasks are defined using reward functions with different tasks having their own optimal agent policy or behavior corresponding to the task reward. RL agents are usually trained for a given task (reward function) or on a distribution of related tasks; most RL agents do not generalize to solving *any* task, even in the same environment. While related machine learning fields like computer vision and natural language processing have shown success at zero-shot (Ramesh et al., 2021) and few-shot (Radford et al., 2021) adaptation to a wide range of downstream tasks, RL lags behind in such functionalities. Unsupervised RL aims to extract reusable information such as skills (Eysenbach et al., 2019; Zahavy et al., 2023), representations (Ghosh et al., 2023; Ma et al., 2023), world-models (Janner et al., 2019; Hafner et al., 2020), or goal-reaching policies (Agarwal et al., 2024; Sikchi et al., 2024a) from the environment using task-independent data to efficiently train RL agents for any task. Recent advances in unsupervised RL (Wu et al., 2019; Touati & Ollivier, 2021a; Blier et al., 2021b; Touati et al., 2023) have shown some promise towards achieving zero-shot RL.

Recently proposed pretraining algorithms (Stooke et al., 2021; Schwarzer et al., 2021b; Sermanet et al., 2017; Nair et al.; Ma et al., 2023) use self-supervised learning to learn representations from large-scale data to facilitate few-shot RL but these representations are dependent on the policies used for collecting the data. These algorithms assume that the large scale data is collected from a "good" policy demonstrating expert task solving behaviors.

Several prior works aim to achieve generalization in multi-task RL by building upon successor features (Dayan, 1993) which represent rewards as a linear combination of state features. These methods have limited generalization capacity to unseen arbitrary tasks. Other works (Mahadevan,

---
*Equal contribution  [1]The University of Texas at Austin  [2]Sony AI. Correspondence to: Siddhant Agarwal <siddhant@cs.utexas.edu>, Harshit Sikchi <hsikchi@utexas.edu>.

*Proceedings of the 42nd International Conference on Machine Learning*, Vancouver, Canada. PMLR 267, 2025. Copyright 2025 by the author(s).

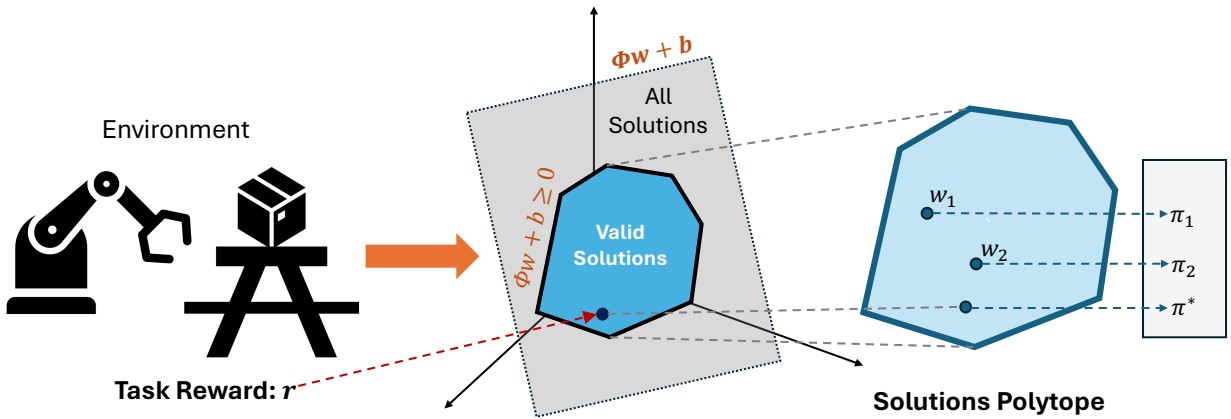

*Figure 1.* **Method Overview:** Visitation distributions corresponding to any policy must obey the Bellman Flow constraint for the dynamical system. This means they must lie on the plane defined by the the Bellman Flow equation. Being a plane, it can be represented using a basis set $\Phi$ and a bias. All valid (non negative) visitation distributions lie within a convex hull on this plane. The boundary of this hull is defined using the non negativity constraints: $\Phi w + b \geq 0$. Each point within this convex hull corresponds to a visitation distribution for a valid policy and is defined simply by the "coordinate" $w$.

2005; Machado et al., 2017; 2018; Bellemare et al., 2019; Farebrother et al., 2023) represent value functions using eigenvectors of the graph Laplacian obtained from a random policy to approximate the global basis of value functions. However, the eigenvectors from a random policy usually cannot represent all value functions. In fact, we show that an alternative strategy of representing visitation distributions using a set of basis functions covers a larger set of solutions than doing the same with value functions. Skill learning methods (Eysenbach et al., 2019; Park et al., 2024b; Eysenbach et al., 2022) view any policy as a combination of skills , but as shown by (Eysenbach et al., 2022), these methods do not recover all possible skills from the MDP. Some recent work has attempted zero-shot RL by decomposing the representation of visitation distributions (Touati & Ollivier, 2021a; Touati et al., 2023), but they learn policy representations as a projection of the reward function which can lead to loss of task-relevant information. We present a stronger, more principled approach for representing any solution of RL in the MDP.

Any policy in the environment can be represented using visitation distributions or the distributions over states and actions that the agent visits when following a policy. We learn a basis set to represent any possible visitation distribution in the underlying environmental dynamics. We draw our inspiration from the linear programming view (Manne, 1960; Denardo, 1970; Nachum & Dai, 2020; Sikchi et al., 2024b) of reinforcement learning; the objective is to find the visitation distribution that maximizes the return (the dot-product of the visitation distribution and the reward) subject to the Bellman Flow constraints. We show that any solution of the Bellman Flow constraint for the visitation distribution can be represented as a linear combination of policy-independent basis functions and a bias. As shown

in Figure 1, any visitation distribution that is a solution of the Bellman Flow for a given dynamical system lies on a plane defined using policy independent basis $\Phi$ and a bias $b$. On this plane, only a small convex region defines the valid (non-negative) visitations distributions. Any visitation distribution in this convex hull can be obtained simply using the "coordinates" $w$. We introduce *Proto-Successor Measure*, the set of basis functions and bias to represent any successor measure (a generalization over visitation distributions) in the MDP that can be learnt using reward-free interaction data. At test time, obtaining the optimal policy reduces to simply finding the linear weights to combine these basis vectors that maximize its dot-product with the user-specified reward. These basis vectors only depend on the state-action transition dynamics of the MDP, independent of the initial state distribution, reward, or policy, and can be thought to compactly represent the entire dynamics.

The contributions of our work are (1) a novel, principled perspective on representation learning for Markov decision processes; (2) an efficient practical instantiation that reduces basis learning to a single-player optimization; and (3) evaluations of a number of tasks demonstrating the capability of our learned representations to quickly infer near-optimal policies.

## 2. Related Work

**Unsupervised Reinforcement Learning:** Unsupervised RL generally refers to a broad class of algorithms that use reward-free data to improve the efficiency of RL algorithms. We focus on methods that learn representations to produce optimal value functions for any given reward function. Representation learning through unsupervised or self-supervised RL has been discussed for both pre-training (Nair et al.; Ma

et al., 2023) and training as auxiliary objectives (Agarwal et al., 2021; Schwarzer et al., 2021a). While using auxiliary objectives for representation learning does accelerate policy learning for downstream tasks, the policy learning begins from scratch for a new task. Pre-training methods like (Ma et al., 2023; Nair et al.) use self-supervised learning techniques from computer vision like masked auto-encoding to learn representations that can be used directly for downstream tasks. These methods use large-scale datasets (Grauman et al., 2022) to learn representations but these are fitted around the policies used for collecting data. These representations do not represent any possible behavior nor are trained to represent Q functions for any reward functions. Several prior works aim to discover intents or skills using a diversity objective. These methods use the fact that the latents or skills should define the output state-visitation distributions thus diversity can be ensured by maximizing mutual information (Warde-Farley et al., 2019; Eysenbach et al., 2019; Achiam et al., 2018; Eysenbach et al., 2022) or minimizing Wasserstein distance (Park et al., 2024b) between the latents and corresponding state-visitation distributions. PSM differs from these works and takes a step towards learning representations optimal for predicting value functions as well as a zero-shot optimal policy for any reward.

**Methods that linearize RL quantities:** Learning basis vectors has been leveraged in RL to allow for transfer to new tasks. Successor features (Barreto et al., 2017) represents rewards as a linear combination of transition features and subsequently the Q-functions are linear in successor features. Several methods have extended successor features (Lehnert & Littman, 2020; Hoang et al., 2021; Alegre et al., 2022; Reinke & Alameda-Pineda, 2021) to learn better policies in more complex domains.

Spectral methods like Proto Value Functions (PVFs) (Mahadevan, 2005; Mahadevan & Maggioni, 2007) instead represent the value functions as a linear combination of basis vectors. It uses the eigenvectors of the random walk operator (graph Laplacian) as the basis vectors. Adversarial Value Functions (Bellemare et al., 2019) and Proto Value Networks (Farebrother et al., 2023) have attempted to scale up this idea in different ways. However, deriving these eigenvectors from a Laplacian is not scalable to larger state spaces. (Wu et al., 2019) recently presented an approximate scalable objective, but the Laplacian is still dependent on random policy which usually makes it incapable of representing all behaviors or Q functions.

Similar to our work, Forward Backward (FB) Representations (Touati & Ollivier, 2021a; Touati et al., 2023) use an inductive bias on the successor measure to decompose it into a forward and backward representation. Unlike FB, our representations are linear on a set of basis features. Additionally, FB ties the reward with the representation of

the optimal policy derived using Q function maximization which can lead to overestimation issues and instability during training as a result of Bellman optimality backups.

## 3. Preliminaries

In this section we introduce some preliminaries and define terminologies that will be used in later sections. We begin with some MDP fundamentals and RL preliminaries followed by a discussion on affine spaces which form the basis for our representation learning paradigm.

### 3.1. Markov Decision Processes

A Markov Decision Process is defined as a tuple $\langle \mathcal{S}, \mathcal{A}, P, r, \gamma, \mu \rangle$ where $\mathcal{S}$ is the state space, $\mathcal{A}$ is the action space, $P : \mathcal{S} \times \mathcal{A} \longmapsto \Delta(\mathcal{S})$ is the transition probability ($\Delta(\cdot)$ denotes a probability distribution over a set), $\gamma \in [0, 1)$ is the discount factor, $\mu$ is the distribution over initial states and $r : \mathcal{S} \times \mathcal{A} \longmapsto \mathbb{R}$ is the reward function. The *task* is specified using the reward function $r$ and the initial state distribution $\mu$. The goal for the RL agent is to learn a policy $\pi_\theta : \mathcal{S} \longmapsto \mathcal{A}$ that maximizes the expected return $J(\pi_\theta) = \mathbb{E}_{s_0 \sim \mu} \mathbb{E}_{\pi_\theta} [\sum_{t=0}^{\infty} \gamma^t r(s_t, a_t)]$.

In this work, we consider a *task-free* MDP which does not provide the reward function or the initial state distribution. Hence, a *task-free* or *reward-free* MDP is simply the tuple $\langle \mathcal{S}, \mathcal{A}, P, \gamma \rangle$. A *task-free* MDP essentially only captures the underlying environment dynamics and can have infinite downstream tasks specified through different reward functions.

The state-action visitation distribution, $d^\pi(s, a)$ is defined as the normalized probability of being in a state $s$ and taking an action $a$ if the agent follows the policy $\pi$ from a state sampled from the initial state distribution. Concretely, $d^\pi(s, a) = (1 - \gamma) \sum_{t=0}^{\infty} \gamma^t \mathbb{P}(s_t = s, a_t = a)$. A more general quantity, successor measure, $M^\pi(s, a, s^+, a^+)$, is defined as the probability of being in state $s^+$ and taking action $a^+$ when starting from the state-action pair $s, a$ and following the policy $\pi$. Mathematically, $M^\pi(s, a, s^+, a^+) = (1 - \gamma) \sum_{t=0}^{\infty} \gamma^t \mathbb{P}(s_t = s^+, a_t = a^+ | s_0 = s, a_0 = a)$. The state-action visitation distribution can be written as $d^\pi(s, a) = \mathbb{E}_{s_0 \sim \mu(s), a_0 \sim \pi(a_0|s_0)}[M^\pi(s_0, a_0, s, a)]$.

Both these quantities, state-action visitation distribution and successor measure, follow the Bellman Flow equations:

$$\sum_a d^\pi(s, a) = (1 - \gamma)\mu(s) +$$

$$\gamma \sum_{s' \in \mathcal{S}, a' \in \mathcal{A}} P(s|s', a') d^\pi(s', a'). \quad (1)$$

For successor measure, the initial state distribution changes

to an identity function

$$\sum_{a^+} M^\pi(s, a, s^+, a^+) = (1-\gamma) \sum_{a^+} \mathbb{1}[s = s^+, a = a^+] + \gamma \sum_{s' \in \mathcal{S}, a' \in \mathcal{A}} P(s^+|s', a') M^\pi(s, a, s', a'). \quad (2)$$

The RL objective has a well studied linear programming interpretation (Manne, 1960). Given any task reward function $r$, the RL objective can be rewritten in the form of a constrained linear program:

$$\max_d \quad \sum_{s,a} d(s,a)r(s,a), \quad s.t. \quad d(s,a) \geq 0 \quad \forall s, a,$$

$$s.t. \quad \sum_a d(s,a) = (1 - \gamma)\mu(s) + \gamma \sum_{s' \in \mathcal{S}, a' \in \mathcal{A}} P(s|s', a')d(s', a')$$

$$(3)$$

and the unique policy corresponding to visitation $d$ is obtained by $\pi(a|s) = \frac{d(s,a)}{\sum_a d(s,a)}$. The Q function can then be defined using successor measure as $Q^\pi(s, a) = \sum_{s^+, a^+} M^\pi(s, a, s+, a+)r(s^+, a^+)$ or $Q^\pi = M^\pi r$. Obtaining the optimal policies requires maximizing the Q function which requires solving $\arg\max_\pi M^\pi r$.

### 3.2. Affine Spaces

Let $\mathcal{V}$ be a vector space and $b$ be a vector. An affine set is defined as $A = b + \mathcal{V} = \{x | x = b + v, v \in \mathcal{V}\}$. Any vector in a vector space can be written as a linear combination of basis vectors, i.e., $v = \sum_i^n \alpha_i v_i$ where $n$ is the dimensionality of the vector space. This property implies that any element of an affine space can be expressed as $x = b + \sum_i^n \alpha_i v_i$. Given a system of linear equations $Ax = c$, with $A$ being an $m \times n$ matrix ($m < n$) and $c \neq 0$, the solution $x$ forms an affine set. Hence, there exists alphas $\alpha_i$ such that $x = b + \sum_i \alpha_i x_i$. The vectors $\{x_i\}$ form the basis set of the null space or *kernel* of $A$. The values $(\alpha_i)$ form the affine coordinates of $x$ for the basis $\{x_i\}$. Hence, for a given system with known $\{x_i\}$ and $b$, any solution can be represented using only the affine coordinates $(\alpha_i)$.

We first explain the theoretical foundations of our method in Section 4 and derive a practical algorithm following the theory in Section 5

## 4. The Basis Set for All Solutions of RL

In this section, we introduce the theoretical results that form the foundation for our representation learning approach. The proof for all the theoretical results can be found in Appendix 8. The goal is to learn policy-independent representations that can represent any valid visitation distribution in the environment (i.e. satisfy the Bellman Flow constraint in Equation 3). With a compact way to represent these dis-

tributions, it is possible to reduce the policy optimization problem to a search in this compact representation space. We will show that state visitation distributions and successor measures form an affine set and thus can be represented as $\sum_i \phi_i w_i^\pi + b$, where $\phi_i$ are basis functions, $w^\pi$ are "coordinates" or weights to linearly combine the basis functions, and $b$ is a bias term. First, we build up the formal intuition for this statement and later we will use a toy example to show how these representations can make policy search easier.

The first constraint in Equation 3 is the Bellman Flow equation. We begin with Lemma 4.1 showing that state visitation distributions that satisfy the Bellman Flow form affine sets.

**Theorem 4.1.** *All possible state-action visitation distributions in an MDP form an affine set.*

While Theorem 4.1 shows that any state-action visitation distribution in an MDP can be written using a linear combination of basis and bias terms, it still depend on the initial state distribution. Moreover, as shown in Equation 1, computing the state-action visitation distribution requires a summation over all states and actions in the MDP which is not always possible. Successor measures are more general than state-action visitation distributions as they encode the visitation of the policy conditioned on a starting state-action pair. Using similar techniques, we show that successor measures also form affine sets.

**Corollary 4.2.** *Any successor measure, $M^\pi$ in an MDP forms an affine set and so can be represented as $\sum_i^d \phi_i w_i^\pi + b$ where $\phi_i$ and $b$ are independent of the policy $\pi$ and $d$ is the dimension of the affine space.*

Following Corollary 4.2, for any $w$, the function $\sum_i^d \phi_i w_i^\pi + b$ will be a solution of Equation 2. Hence, given $\Phi$ ($\phi_i$ stacked together) and $b$, we do not need the second constraint on the linear program (in Equation 3) anymore. The other constraint: $\phi_i w_i + b \geq 0$ still remains which $w$ needs to satisfy. We discuss ways to manage this constraint in Section 5.3. The linear program given a reward function now becomes,

$$\max_w \quad \mathbb{E}_\mu[(\Phi w + b)r]$$

$$s.t. \quad \Phi w + b \geq 0 \quad \forall s, a. \quad (4)$$

In fact, any visitation distribution that is a policy-independent linear transformation of $M^\pi$, such as state visitation distribution or future state-visitation distribution, can be represented in the same way as shown in Corollary 4.3.

**Corollary 4.3.** *Any quantity that is a policy-independent linear transformation of $M^\pi$ can be written as a linear combination of policy-independent basis and bias terms.*

**Extension to Continuous Spaces:** In continuous spaces, the basis matrices $\phi$ and bias $b$ become functions $\phi : S \times A \times S \to \mathbb{R}^d$ and $b : S \times A \times S \to \mathbb{R}$. The linear equation with matrix operations becomes a linear equation with functional

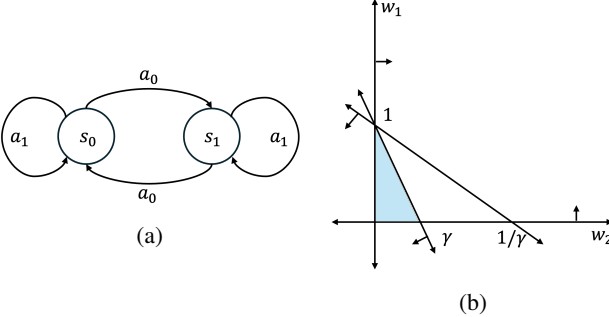

(a)

(b)

*Figure 2.* (left) A Toy MDP with 2 states and 2 actions to depict how the linear program of RL is reduced using precomputation. (right) The corresponding simplex for $w$ assuming the initial state distribution is $\mu = (1, 0)^T$.

transformations, and any sum over states is replaced with expectation under the data distribution.

**Toy Example:** Let's consider a simple 2 state MDP (as shown in Figure 2a) to depict how the precomputation and inference will take place. Consider the state-action visitation distribution as in Equation 1. For this simple MDP, the $\Phi$ and $b$ can be computed using simple algebraic manipulations. For a given initial state-visitation distribution, $\mu$ and $\gamma$, the state-action visitation distribution $d = (d(s_0, a_0), d(s_1, a_0), d(s_0, a_1), d(s_1, a_1))^T$ can be written as,

$$d = w_1 \begin{pmatrix} \frac{-\gamma}{1+\gamma} \\ \frac{-1}{1+\gamma} \\ 1 \\ 0 \end{pmatrix} + w_2 \begin{pmatrix} \frac{-1}{1+\gamma} \\ \frac{-\gamma}{1+\gamma} \\ 0 \\ 1 \end{pmatrix} + \begin{pmatrix} \frac{\mu(s_0)+\gamma\mu(s_1)}{1+\gamma} \\ \frac{\mu(s_1)+\gamma\mu(s_0)}{1+\gamma} \\ 0 \\ 0 \end{pmatrix}. \quad (5)$$

The derivation for these basis vectors and the bias vector can be found in Appendix A.6. Equation 5 represents any vector that is a solution of Equation 1 for the simple MDP. Any state-action visitation distribution possible in the MDP can now be represented using only $w = (w_1, w_2)^T$. The only constraint in the linear program of Equation 4 is $\Phi w + b \geq 0$. Looking closely, this constraint gives rise to four inequalities in $w$ and the linear program reduces to,

$$\max_{w_1, w_2} \quad (\frac{-\gamma w_1 - w_2}{1 + \gamma}, \frac{-w_1 - \gamma w_2}{1 + \gamma}, w_1, w_2)^T r$$
$$s.t. \quad w_1 + \gamma w_2 \leq \mu(s_0) + \gamma\mu(s_1) \quad . \quad (6)$$
$$\gamma w_1 + w_2 \leq \mu(s_1) + \gamma\mu(s_0)$$
$$w_1 \geq 0, w_2 \geq 0$$

The inequalities in $w$ give rise to a simplex as shown in Figure 2b. For any specific instantiation of $\mu$ and $r$, the optimal policy can be easily found. For instance, if $\mu = (1, 0)^T$ and the reward function, $r = (1, 0, 1, 0)^T$, the optimal $w$ will be obtained at the vertex ($w_1 = 1, w_2 = 0$) and the corresponding state-action visitation distribution is $d = (0, 0, 1, 0)^T$. As shown for the toy MDP, the successor measures form a simplex as discussed in (Eysenbach et al., 2022). Spectral Methods following Proto Value Functions (Mahadevan & Maggioni, 2007) have instead tried to learn policy indepen-

dent basis functions, $\Phi^{vf}$ to represent value functions as a linear span, $V^\pi = \Phi^{vf}w^\pi$. Some prior works (Dadashi et al., 2019) have already argued that value functions do not form convex polytopes. We show through Theorem 4.4 that for identical dimensionalities, the span of value functions using basis functions represent a smaller class of value functions than the set of value functions that can be represented using the span of the successor measure.

**Theorem 4.4.** *Given a $d$-dimensional basis $\mathbf{B} : \mathbb{R}^n \to \mathbb{R}^d$, define $span\{\mathbf{B}\}$ as the span of all linear combinations of basis $\mathbf{B}$. Further define $span\{\mathbf{B}r\}$ as the span of inner products of all linear combinations of basis $\mathbf{B}$ and all possible reward functions $r$. Let $span\{\Phi^{vf}\}$ denote the space of the value functions spanned by $\Phi^{vf}$ while $\{span\{\Phi\}r\}$ denotes the space of value functions using the successor measures spanned by $\Phi$. For the same dimensionality of task (policy or reward) independent basis, $span\{\Phi^{vf}\} \subseteq \{span\{\Phi\}r\}$ for some $\Phi$.*

Approaches such as Forward Backward Representations (Touati & Ollivier, 2021a) have also been based on representing successor measures but they force a latent variable $z$ representing the policy to be a function of the reward for which the policy is optimal. The linear one-to-one mapping between reward functions and corresponding policies are incorrect as there can be many rewards leading to the same optimal policy as well as many optimal policies corresponding to a single reward function. In addition, the forward map that they propose is a function of this latent $z$. We, on the other hand, propose a representation that is truly independent of the policy or the reward.

## 5. Method

In this section, we start by introducing the core practical algorithm for representation learning inspired by the theory discussed in Section 4 for obtaining $\Phi$ and $b$. We then discuss the inference step, i.e., obtaining $w$ for a given reward function.

### 5.1. Learning $\Phi$ and $b$

For a given policy $\pi$, its successor measure under our framework is denoted by $M^\pi = \Phi w^\pi + b$ with $w^\pi$ the only object depending on policy. Given an offline dataset with density $\rho$, we follow prior works (Touati & Ollivier, 2021a; Blier et al., 2021b) and model densities $m^\pi = M^\pi/\rho$ learned with the following objective:

$$L^\pi(\Phi, b, w^\pi) = -(1-\gamma)\mathbb{E}_{s,a\sim\rho}[m^{\Phi,b,w^\pi}(s, a, s, a)]$$
$$+ \frac{1}{2}\mathbb{E}_{s,a,s'\sim\rho,s^+,a^+\sim\rho}[(m^{\Phi,b,w^\pi}(s, a, s^+, a^+) -$$
$$\gamma\bar{m}^{\bar{\Phi},\bar{b},\bar{w}^\pi}(s', \pi(s'), s^+, a^+))^2]. \quad (7)$$

The above objective only requires samples $(s, a, s')$ from the reward-free dataset and a random state-action pair $(s^+, a^+)$

(also sampled from the same data) to compute $L(\pi)$.

A $\Phi$ and $b$ that allows for minimizing the $L(\pi)$ for all $\pi \in \Pi$ forms a solution to our representation learning problem. But how do we go about learning such $\Phi$ and $b$? A naïve way to implement learning $\Phi$ and $b$ is via a bi-level optimization. We sample policies from the policy space of $\Pi$, for each policy we learn a $w^\pi$ that optimizes the policy evaluation loss (Eq 7) and take a gradient update w.r.t $\Phi$ and $b$. In general, the objective can be optimized by any two-player game solving strategies with $[\Phi, b]$ as the first player and $w^\pi$ as the second player. Instead, in the next section, we present an approach to simplify learning representations to a single-player game.

### 5.2. Simplifying Optimization via a Discrete Codebook of Policies

Learning a new $w^\pi$ for each specific sampled policy $\pi$ does not leverage precomputations and requires retraining from scratch. We propose parameterizing $w$ to be conditional on policy, which allows leveraging generalization between policies that induce similar visitation and as we show, will allow us to simplify the two player game into a single player optimization. In general, policies are high-dimensional objects and compressing them can result in additional overhead. Compression by parameterizing policies with a latent variable $z$ is another alternative but presents the challenge of covering the space of all possible policies by sampling $z$. Instead, we propose using a discrete codebook of policies as a way to simulate uniform sampling of all possible policies with support in the offline dataset.

**Discrete Codebook of Policies**: Denote $z$ as a compact representation of policies. We propose to represent $z$ as a random sampling *seed* that will generate a deterministic policy from the set of supported policies as Equation 8. In other words, $z$ will be a random integer ($z \leq 2^h$ (represented using $h$ bits) that will represent a deterministic policy.

$$\pi(a|s, z) = \text{Uniform Sample}(\text{seed} = z + \text{hash}(s)). \quad (8)$$

The above sampling strategy defines a unique mapping from a seed to a policy. If the seed generator is unbiased, the approach provably samples from among all possible deterministic policies uniformly. Now, with policy $\pi_z$ and $w_z$ parameterized as a function of $z$ we derive the following single-player reduction to learn $\Phi, b, w$ jointly.

$$\texttt{PSM-objective:} \quad \underset{\Phi, b, w(z)}{\arg\min} \mathbb{E}_z[L^{\pi_z}(\Phi, b, w(z))]. \quad (9)$$

The disentanglement between the policies sampled during training and the corresponding reward makes Equation 9 more stable than similar looking objectives like FB (Touati & Ollivier, 2021b). Equation 9 is absent of any policy learning via maximization of a moving reward function. In particular, PSM uses a relatively mild form of off-policy learning

that is more stable than the one derived by maximization (Farebrother et al., 2023).

### 5.3. Fast Inference on Downstream Tasks

After obtaining $\Phi$ and $b$ via the pretraining step, the only parameter to compute for obtaining the optimal Q function for a downstream task in the MDP is $w$. As discussed earlier, $Q^* = \max_w(\Phi w + b)r$ but simply maximizing this objective will not yield a Q function. The linear program still has a constraint of $\Phi w + b \geq 0, \forall s, a$. We solve the constrained linear program by constructing the Lagrangian dual using Lagrange multipliers $\lambda(s, a)$. The dual problem is shown in Equation 10. Here, we write the corresponding loss for the constraint as $\min(\Phi w + b, 0)$.

$$\max_{\lambda \geq 0} \min_w -\Phi wr - \sum_{s,a} \lambda(s, a) \min(\Phi w + b, 0). \quad (10)$$

Once $w^*$ is obtained, the corresponding $M^*$ and $Q^*$ can be easily computed. The policy can be obtained as $\pi^* = \arg\max_a Q^*(s, a)$ for discrete action spaces and via DDPG style policy learning for continuous action spaces.

## 6. Connections to Successor Features

In this section, we uncover the theoretical connections between PSM and successor features. Successor Features (Barreto et al., 2017) ($\psi^\pi(s, a)$) are defined as the discounted sum of state features $\varphi(s)$, $\psi^\pi(s, a) = \mathbb{E}_\pi[\sum_t \gamma^t \varphi(s_t)]$. These state features can be used to span reward functions as $r = \varphi z$. Using this construction, the Q function is linear in $z$ as $Q(s, a) = \psi^\pi(s, a)z$. We can establish a simple relation between $M^\pi$ and $\psi^\pi$, $\psi^\pi(s, a) = \int_{s'} M^\pi(s, a, s')\varphi(s')ds'$. This connection shows that, like successor measures, successor features can also be represented using a similar basis.

**Theorem 6.1.** *Successor Features $\psi^\pi(s, a)$ belong to an affine set and can be represented using a linear combination of basis functions and a bias.*

Interestingly, instead of learning the basis of successor measures, we show below that PSM can also be used to learn the basis of successor features. While traditional successor feature-based methods assume that the state features $\varphi$ are provided, PSM can be used to jointly learn the successor feature and the state feature. We begin by introducing the following Lemma 6.2 from (Touati et al., 2023) which connects an a specific decomposition for successor measures to the ability of jointly learning state features and successor representations,

**Lemma 6.2.** *(Theorem 13 of (Touati et al., 2023)) For an offline dataset with density $\rho$, if the successor measure is represented as $M^\pi(s, a, s^+) = \psi^\pi(s, a)\varphi(s^+)\rho(s^+)$, then $\psi$ is the successor feature $\psi^\pi(s, a)$ for state feature $\varphi(s)^T(\mathbb{E}_\rho(\varphi\varphi^T))^{-1}$.*

According to Lemma 6.2, if $M^\pi(s, a, s^+) = \psi^\pi(s, a)\varphi(s^+)\rho(s^+)$, then the corresponding successor feature is $\psi^\pi(s, a)$ and the state feature is $\varphi(s)^T(\mathbb{E}_\rho(\varphi\varphi^T))^{-1}$. PSM represents successor measures as $M^\pi(s, a, s^+) = \phi(s, a, s^+)w^\pi\rho(s^+)$ (for simplicity, combining the bias within the basis without loss of generality). It can be shown that if the basis learned for successor measure using PSM, $\phi(s, a, s^+)$ is represented as a decomposition $\phi_\psi(s, a)^T\varphi(s^+)$, $\phi_\psi(s, a)$ forms the basis for successor features for the state features $\varphi(s)^T(\mathbb{E}_\rho(\varphi\varphi^T))^{-1}$. Formally, we present the following theorem,

**Theorem 6.3.** *For the PSM representation $M^\pi(s, a, s^+) = \phi(s, a, s^+)w^\pi$ and $\phi(s, a, s^+) = \phi_\psi(s, a)^T\varphi(s^+)$, the successor feature $\psi^\pi(s, a) = \phi_\psi(s, a)w^\pi$ for the state feature $\varphi(s)^T(\mathbb{E}_\rho(\varphi\varphi^T))^{-1}$.*

Thus, successor features can be obtained by enforcing a particular inductive bias to decompose $\phi$ in PSM. For rewards linear in state features ($r(s) = \langle\varphi(s) \cdot z\rangle$ for some weights $z$), the $Q$-functions remain linear given by $Q^\pi(s, a) = \phi_\psi(s, a)w^\pi\mathbb{E}_\rho[\varphi(s)z]$. A natural question to ask is, with this decomposition, do we lose the expressibility of PSM compared to the methods that compute basis spanning value functions, thus contradicting Theorem 4.4? The answer is negative, since (1) even though the value function seems to be linear combination of some basis with weights $w^\pi$, these weights are not tied to $z$ or the reward. The relationship between the optimal weights $w^{\pi^*}$ and $z$ defining the reward function is not necessarily linear as the prior works assume, and (2) the decomposition $\phi(s, a, s^+) = \phi_\psi(s, a)\varphi(s^+)$ reduces the representation capacity of the basis. While prior works are only able to recover features pertaining to this reduced representation capacity, PSM does not assume this decomposition and can learn a larger representation space. Additionally, a number of SF methods assume access to these features or learn them using an auxiliary objective, assuming that the obtained features would be sufficient to span successor measures. On the other hand, PSM is able to extract the features relevant for learning $M^\pi$ for all $\pi$.

# 7. Experimental Study

Our experiments evaluate how PSM can be used to encapsulate a *task-free* MDP into a representation that will enable zero-shot inference on any downstream task. In the experiments we investigate a) the quality of value functions learned by PSM (Section 7.2), b) the zero-shot performance of PSM in contrast to other baselines on discrete tasks (Section 7.1 and Appendix C.2), c) the ability to learn general goal-reaching skills arising from the PSM objective on a robot manipulation task (Section7.2)and finally d) Suitability of learned PSM representations for enabling zero-shot RL in continuous state-action space tasks. (Section 7.3)

**Baselines:** We compare against methods that are commonly used and are the state of the art in spanning the space of reward functions: Laplacian features (Wu et al., 2018), Forward-Backward (Touati et al., 2023) and HILP (Park et al., 2024a). Laplacian features learn features of a state by considering eigenvectors of a graph Laplacian induced by a random walk. These features $\psi(s) \in \mathbb{R}^d$ obtained for each state are used to define a reward function conditioned on a reward $r(s; \psi) = \psi(s) \cdot z$ where $z$ is sampled uniformly from a unit d-dimensional sphere. For each $z$ an optimal policy is pretrained from the dataset on the induced reward function. During inference the corresponding $z$ for a given reward function is obtained as a solution to the following linear regression: $\min_z \mathbb{E}_s[(\psi^\top \cdot z - r(s))^2]$. Similar to Laplacian features, we consider a couple other features in the SF framework, one that uses one-step forward dynamics predictability (method named "FDM") and one that uses SVD decomposition of successor representation (method named "SVD"). Forward-backward (FB) learns both the optimal policy and state features jointly for all reward that are in the linear span of state-features. FB methods typically assume a goal-conditioned prior during pretraining which typically helps in learning policies that reach various states in the dataset. HILP (Park et al., 2024a) makes two changes to FB: a) Reduces the tasks to be goal reaching and b) Uses a more performant offline RL method, IQL (Kostrikov et al., 2021) to learn features. We provide detailed experimental setup and hyperparameters in Appendix B.3.

## 7.1. Zero shot Value function and Optimal Policy prediction

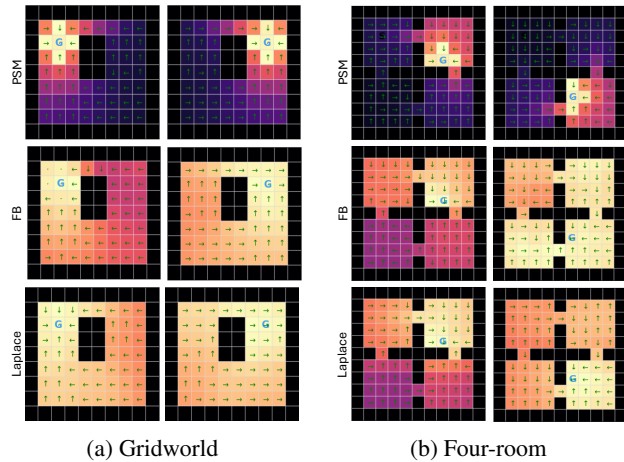

(a) Gridworld       (b) Four-room

*Figure 3.* **Qualitative results on a gridworld and four-room:** G denotes the goal sampled for every episode. The black regions are the boundaries/obstacles. The agent needs to navigate across the grid and through the small opening (in case of four-room) to reach the goal. We visualize the optimal Q-functions inferred at test time for the given goal in the image. The arrows denote the optimal policy. (Top row) Results for PSM, (Middle Row) Results for FB, (Bottom row) Results for Laplacian Eigenfunctions.

| | Task | Laplace | FDM | SVD | FB | HILP | PSM |
|---|---|---|---|---|---|---|---|
| **Walker** | Stand | $243.70 \pm 151.40$ | $834.42 \pm 39.36$ | $822.52 \pm 99.25$ | $902.63 \pm 38.94$ | $607.07 \pm 165.28$ | $872.61 \pm 38.81$ |
| | Run | $63.65 \pm 31.02$ | $269.27 \pm 27.52$ | $332.06 \pm 12.87$ | $392.76 \pm 31.29$ | $107.84 \pm 34.24$ | $351.50 \pm 19.46$ |
| | Walk | $190.53 \pm 168.45$ | $647.76 \pm 180.99$ | $852.06 \pm 13.67$ | $877.10 \pm 81.05$ | $399.67 \pm 39.31$ | $891.44 \pm 46.81$ |
| | Flip | $48.73 \pm 17.66$ | $441.09 \pm 66.48$ | $444.43 \pm 106.56$ | $206.22 \pm 162.27$ | $277.95 \pm 59.63$ | $640.75 \pm 31.88$ |
| | **Average(*)** | 136.65 | 548.14 | 612.76 | 594.67 | 348.13 | **689.07** |
| **Cheetah** | Run | $96.32 \pm 35.69$ | $161.90 \pm 63.51$ | $171.55 \pm 15.05$ | $257.59 \pm 58.51$ | $68.22 \pm 47.08$ | $244.38 \pm 80.00$ |
| | Run Backward | $106.38 \pm 29.4$ | $280.16 \pm 17.76$ | $199.79 \pm 28.05$ | $307.07 \pm 14.91$ | $37.99 \pm 25.16$ | $296.44 \pm 20.14$ |
| | Walk | $409.15 \pm 56.08$ | $589.85 \pm 177.14$ | $672.03 \pm 115.86$ | $799.83 \pm 67.51$ | $318.30 \pm 168.42$ | $984.21 \pm 0.49$ |
| | Walk Backward | $654.29 \pm 219.81$ | $945.91 \pm 63.13$ | $858.33 \pm 42.89$ | $980.76 \pm 2.32$ | $349.61 \pm 236.29$ | $979.01 \pm 7.73$ |
| | **Average(*)** | 316.53 | 494.46 | 475.42 | 586.31 | 193.53 | **626.01** |
| **Quadruped** | Stand | $854.50 \pm 41.47$ | $940.01 \pm 29.69$ | $894.30 \pm 35.79$ | $740.05 \pm 107.15$ | $409.54 \pm 97.59$ | $842.86 \pm 82.18$ |
| | Run | $412.98 \pm 54.03$ | $434.08 \pm 21.85$ | $433.66 \pm 21.89$ | $386.67 \pm 32.53$ | $205.44 \pm 47.89$ | $431.77 \pm 44.69$ |
| | Walk | $494.56 \pm 62.49$ | $459.43 \pm 32.44$ | $457.74 \pm 64.93$ | $566.57 \pm 53.22$ | $218.54 \pm 86.67$ | $603.97 \pm 73.67$ |
| | Jump | $642.84 \pm 114.15$ | $709.07 \pm 71.99$ | $687.75 \pm 40.59$ | $581.28 \pm 107.38$ | $325.51 \pm 93.06$ | $596.37 \pm 94.23$ |
| | **Average(*)** | 601.22 | **635.65** | **618.36** | 568.64 | 289.75 | **618.74** |
| **Pointmass** | Top Left | $713.46 \pm 58.90$ | $891.56 \pm 49.16$ | $853.79 \pm 35.49$ | $897.83 \pm 35.79$ | $944.46 \pm 12.94$ | $831.43 \pm 69.51$ |
| | Top Right | $581.14 \pm 214.79$ | $379.75 \pm 84.38$ | $442.92 \pm 123.84$ | $274.95 \pm 197.90$ | $96.04 \pm 166.34$ | $730.27 \pm 58.10$ |
| | Bottom Left | $689.05 \pm 37.08$ | $563.53 \pm 213.60$ | $517.19 \pm 142.36$ | $517.23 \pm 302.63$ | $192.34 \pm 177.48$ | $451.38 \pm 73.46$ |
| | Bottom Right | $21.29 \pm 42.54$ | $10.96 \pm 13.51$ | $57.74 \pm 80.87$ | $19.37 \pm 33.54$ | $0.17 \pm 0.29$ | $43.29 \pm 38.40$ |
| | **Average(*)** | **501.23** | 461.45 | 467.91 | 427.34 | 308.25 | **514.09** |

*Table 1.* Table shows comparison (over 5 seeds) of zero-shot RL performance between different methods with representation size of $d = 128$. PSM demonstrates a marked improvement over prior methods. (*) denotes statistically significant through Mann-Whitney U Test with level 0.05.

In this section, we consider goal-conditioned rewards on a discrete gridworld and the classic four-room environments.

**Task Setup:** Both environments have discrete state and action spaces. The action space consists of *five* actions: $\{up, right, down, left, stay\}$. We collect transitions in the environment by uniformly spawning the agent and taking a random-uniform action. This allows us to form our of-fline reward-free dataset will full coverage to train $\Phi$ and $b$. During inference, we sample a goal and infer the optimal Q function on the goal. Since the reward function is given by $r(s) = \mathbb{1}_{s=g}$, the inference looks like $Q(s, a) = \max_w \Phi(s, a, g)w \quad s.t. \quad \Phi(s, a, s')w + b(s, a, s') \geq 0 \quad \forall s, a, s'$. Figure 3 shows the Q function and the corresponding optimal policy (when executed from a fixed start state) on the gridworld and the four-room environment. As illustrated clearly, for both the environments, the optimal Q function and policy can be obtained zero-shot for any given goal-conditioned downstream task. The error rate of each of the method on these tasks are presented in Appendix C.2

**Comparison to baselines:** We can draw a couple of conclusions from the visualization of the Q functions inferred by the different methods. First, the Q function learnt by PSM is more sharply concentrated on optimal state-action pairs compared to the two baselines. Both baselines have more uniform value estimates, leaving only a minor differential over state values. Secondly, the baselines produce far more incorrect optimal actions (represented by the green arrows) compared to PSM.

### 7.2. Learning zero-shot policies for manipulation

We consider the Fetch-Reach environment (Figure 4) with continuous states and discrete actions (Touati & Ollivier, 2021a). A dataset of size 1M is constructed using DQN+RND. FB, Laplacian and PSM all use this dataset to learn pretrained objects that can be used for zero-shot RL.

We observe that PSM outperforms baselines FB and Laplacian in its ability to learn a zero-shot policy. One key observation is that PSM learning is stable whereas FB exhibits a drop in performance, likely due to the use of Bellman optimality backups resulting in overestimation bias during training. Laplacian's capacity to output zero-shot policies is far exceeded by PSM because Laplacian methods construct the graph Laplacian for random policies and may not be able to represent optimal value functions for all rewards.

### 7.3. Learning Zero-shot Policies for Continuous Control

We use the ExoRL suite (Yarats et al., 2022) for obtaining exploratory datasets collected by running RND (Burda et al., 2019). PSM objective in Equation 9 directly enables learning the basis for successor measures. We decompose the basis representation $\phi(s, a, s^+)$ to $\phi_\psi(s, a)^T \varphi(s^+)$ as discussed in detail in Section 6. PSM thus ensures that $\varphi(s^+)$ can be used to construct basic features to span any reward function. Note that this is not a limiting assumption, as the features can be arbitrarily non-linear in states. In these experiments, we compare the ability of PSM to obtain these representations as compared to prior zero-shot RL methods. Experimental details can be found in Appendix B.3.

Table 1 compares PSM's zero-shot performance in contin-

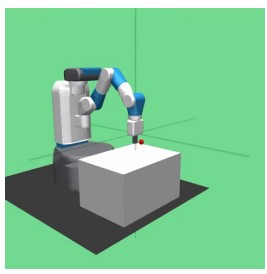

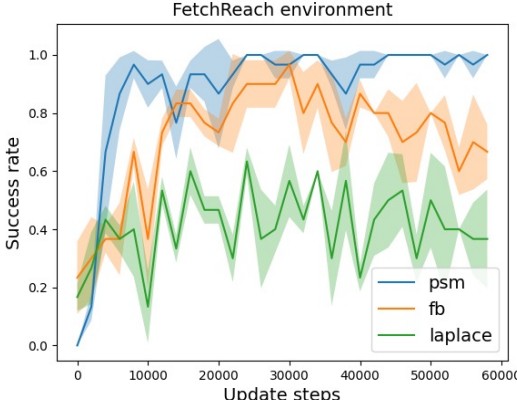

*Figure 4.* **Quantitative results on FetchReach:** The success rates (averaged over 3 seeds) are plotted (along with the standard deviation as shaded) with respect to the training updates for PSM, FB and Laplacian. PSM quickly reaches optimal performance while FB shows instability in maintaining its optimality. Laplacian is far from the optimal performance.

uous state-action spaces to representative methods - Laplacian, FB, and HILP. We note that to make the comparisons fair, we use the same representation dimension of $d = 128$, the same discount factor, and the same inference and policy extraction across environments for a particular method. Overall, PSM performs consistently better or is competitive to baselines across the environments. Ablations studying effect of latent dimensionality can be found in Appendix C.

## 8. Conclusion

This paper introduces Proto Successor Measures (PSM), a zero-shot RL method that compresses any MDP to allow for optimal policy inference *for any reward function* without additional environmental interactions. This framework marks a step in the direction of moving away from common ideology in RL to solve single tasks optimally, and rather pretraining reward-free agents that are able to solve an infinite number of tasks. PSM is based on the principle that successor measures are solutions to an affine set and proposes an efficient and mathematically grounded algorithm to extract the basis for the affine set. Our empirical results show that PSM can produce the optimal Q function and the optimal policy for a number of goal-conditioned as well as reward-specified tasks in a number of environments

outperforming prior baselines.

**Limitations and Future Work:** PSM shows that any MDP can be compressed to a representation space that allows zero-shot RL, but it remains unclear as to what the size of the representation space should be. A large representational dimension can lead to increased compute requirements and training time with a possible chance of overfitting, and a small representation dimension can fail to capture nuances about environments that have non-smooth environmental dynamics. An interesting future direction would be to study the impact of dataset coverage on zero-shot RL performance.

## Acknowledgements

The authors would like to thank Scott Niekum, Ahmed Touati, Ishan Durugkar and Ben Eysenbach for inspiring discussions during this work. We thank Caleb Chuck, Max Rudolph, and members of MIDI Lab for their feedback on this work. SA, HS, and AZ are supported by NSF 2340651 and ARO W911NF-24-1-0193. This work has in part taken place in the Learning Agents Research Group (LARG) at the Artificial Intelligence Laboratory, The University of Texas at Austin. LARG research is supported in part by the National Science Foundation (FAIN-2019844, NRT-2125858), the Office of Naval Research (N00014-18-2243), Army Research Office (W911NF-23-2-0004, W911NF-17-2-0181), DARPA (Cooperative Agreement HR00112520004 on Ad Hoc Teamwork), Lockheed Martin, and Good Systems, a research grand challenge at the University of Texas at Austin. The views and conclusions contained in this document are those of the authors alone. Peter Stone serves as the Executive Director of Sony AI America and receives financial compensation for this work. The terms of this arrangement have been reviewed and approved by the University of Texas at Austin in accordance with its policy on objectivity in research.

## Impact Statement

This paper presents work whose goal is to advance the field of Machine Learning. There are many potential societal consequences of our work, none which we feel must be specifically highlighted here.

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

# Appendix

# A. Theoretical Results

In this section, we will present the proofs for all the Theorems and Corollaries stated in Section 4 and 6.

## A.1. Proof of Theorem 4.1

*Theorem* 4.1. All possible state-action visitation distributions in an MDP form an affine set.

*Proof.* Any state-action visitation distribution, $d^\pi(s, a)$ must satsify the Bellman Flow equation:

$$\sum_a d^\pi(s, a) = (1 - \gamma)\mu(s) + \gamma \sum_{s',a'} \mathbb{P}(s|s', a')d^\pi(s', a'). \tag{11}$$

This equation can be written in matrix notation as:

$$\sum_a d^\pi = (1 - \gamma)\mu + \gamma P^T d^\pi. \tag{12}$$

Rearranging the terms,

$$(S - \gamma P^T)d^\pi = (1 - \gamma)\mu, \tag{13}$$

where $S$ is the matrix for $\sum_a$ of size $|\mathcal{S}| \times |\mathcal{S}||\mathcal{A}|$ with only $|\mathcal{A}|$ entries set to 1 corresponding to the state denoted by the row. This equation is an affine equation of the form $Ax = b$ whose solution set forms an affine set. Hence all state-visitation distributions $d^\pi$ form an affine set.

In the continuous spaces, the visitation distributions would be represented as functions: $d^\pi : S \times A \to \mathbb{R}$ rather than vectors in $[0, 1]^{S \times A}$. The state-action visitation distribution $d^\pi(s, a)$ will satisfy the following continuous Bellman Flow Equation,

$$\int_A d^\pi(s, a)da = (1 - \gamma)\mu(s) + \gamma \int_S \int_A \mathbb{P}(s|s', a')d^\pi(s', a')ds'da'. \tag{14}$$

This equation is the same as Equation 11 except, the vectors representing distributions are replaced by functions and the discrete operator $\sum$ is replaced by $\int$.

The Bellman Flow operator can be defined as $T$ that acts on $d^\pi$ as,

$$T[d^\pi](s) = \int_A d^\pi(s, a)da - \gamma \int_S \int_A \mathbb{P}(s|s', a')d^\pi(s', a')ds'da'. \tag{15}$$

From Equation 14, $T[d^\pi](s) = (1 - \gamma)\mu(s)$. The operator $T$ is a linear operator, hence $d^\pi(s, a)$ forms an affine space.

$\square$

## A.2. Proof of Corollary 4.2

*Corollary* 4.2. Any successor measure, $M^\pi$, in an MDP forms an affine set and so can be represented as $\sum_i^d \phi_i w_i^\pi + b$ where $\phi_i$ and $b$ are independent of the policy $\pi$ and $d$ is the dimension of the affine space.

*Proof.* Using Theorem 4.1, we have shown that state-action visitation distributions form affine sets. Similarly, successor measures, $M^\pi(s, a, s^+, a^+)$ are solutions of the Bellman Flow equation:

$$M^\pi(s, a, s^+, a^+) = (1 - \gamma)\mathbb{1}[s = s^+, a = a^+] + \gamma \sum_{s',a' \in \mathcal{S}\mathcal{A}} P(s^+|s', a')M^\pi(s, a, s', a')\pi(a^+|s^+). \tag{16}$$

Taking summation over $a^+$ on both sides gives us an equation very similar to Equation 11 and so can be written by rearranging as,

$$(S - \gamma P^T)M^\pi = (1 - \gamma)\mathbb{1}[s = s^+]. \tag{17}$$

With similar arguments as in Lemma 4.1, $M^\pi$ also forms an affine set.

Following the previous proof, in continuous spaces, $M^\pi$ becomes a function $M^\pi : S \times A \times S \times A \to \mathbb{R}$ and the Bellman Flow equation transforms to,

$$M^\pi(s, a, s^+, a^+) = (1 - \gamma)p(s = s^+, a = a^+) + \gamma \int_S \int_A P(s^+|s', a')M^\pi(s, a, s', a')\pi(a^+|s^+)ds'da'. \tag{18}$$

Integrating both sides over $a^+$, the Bellman Flow operator $T$ can be constructed that acts on $M^\pi$,

$$T[M^\pi](s, a, s^+) = \int_A M^\pi(s, a, s^+, a^+)da^+ - \gamma \int_S \int_A P(s^+|s', a')M^\pi(s, a, s', a')ds'da' \tag{19}$$

$$\implies T[M^\pi](s, a, s^+) = (1 - \gamma)p(s = s^+, a = a^+) \tag{20}$$

As $T$ is a linear operator, $M^\pi$ belongs to an affine set.

Any element $x$ of an affine set of dimensionality $d$, can be written as $\sum_i^d \phi_i w_i + b$ where $\langle \phi_i \rangle$ are the basis and $b$ is a bias vector. The basis is given by the null space of the matrix operator $(S - \gamma P^T)$ ($T$ in case of continuous spaces). Since the operator $(S - \gamma P^T)$ (and $T$) and the vector $(1 - \gamma)\mathbb{1}[s = s^+]$ (and function $(1 - \gamma)p(s = s^+, a = a^+)$) are independent of the policy, the basis $\Phi$ and the bias $b$ are also independent of the policy. $\qquad \square$

### A.3. Proof of Theorem 4.4

*Theorem* 4.4. Given a $d$-dimensional basis $\mathbf{B} : \mathbb{R}^n \to \mathbb{R}^d$, define $span\{\mathbf{B}\}$ as the span of all linear combinations of basis $\mathbf{B}$. Further define $span\{\mathbf{B}r\}$ as the span of inner products of all linear combinations of basis $\mathbf{B}$ and all possible reward functions $r$. Let $span\{\Phi^{vf}\}$ denote the space of the value functions spanned by $\Phi^{vf}$ while $\{span\{\Phi\}r\}$ denotes the space of value functions using the successor measures spanned by $\Phi$. For the same dimensionality of task (policy or reward) independent basis, $span\{\Phi^{vf}\} \subseteq \{span\{\Phi\}r\}$ for some $\Phi$.

*Proof.* We need to show that any element that belongs to the set $span\{\Phi^{vf}\}$ also belongs to the set $\{span\{\Phi\}r\}$.

Any element belonging to the set $\{span\{\Phi^{vf}\}\}$ is represented by,

$$V^\pi(s) = \sum_i \beta_i^\pi \Phi_i^{vf}(s).$$

$\beta_i^\pi$ can be written as $\frac{\beta_i^\pi}{k_i}k_i$ where $k_i = \sum_{s'} r(s')$ for some $r$. This means,

$$V^\pi(s) = \sum_i \left[\frac{\beta_i^\pi}{k_i} \sum_{s'} r(s')\right] \Phi_i^{vf}(s)$$

$$= \sum_i w_i^\pi \sum_{s'} \Phi_i(s, s')r(s')$$

where $w_i^\pi = \frac{\beta_i^\pi}{k_i}$, $\Phi_i(s, s') = \Phi_i^{vf}(s)\mathbb{1}_{s=s'}$. This implies, for every instance of $V^\pi \in span\{\Phi^{vf}\}$, there exists some instance in $\{span\{\Phi\}r\}$ for some $\Phi$ and $r$.

Lets see when an element in $\{span\{\Phi\}r\}$ belongs in $span\{\Phi^{vf}\}$. We start from an element belonging to set $\{span\{\Phi\}r\}$ as represented by,

$$V^\pi(s) = \sum_i w_i^\pi \sum_{s'} \Phi_i(s, s')r(s')$$

If we assume a special $\Phi_i(s, s') = \sigma_i(s)\eta_i(s')$,

$$V^\pi(s) = \sum_i w_i^\pi \sum_{s'} \Phi_i(s, s')r(s')$$

$$= \sum_i \left[w_i^\pi \sum_{s'} \eta_i(s')r(s')\right]\sigma_i(s)$$

$$= \sum_i \beta_i^\pi \Phi_i^{vf}(s)$$

where $\beta_i^\pi = \left[w_i^\pi \sum_{s'} \eta_i(s')r(s')\right]$ and $\Phi^{vf}(s) = \sigma_i(s)$. This implies for only for the special case $\Phi_i(s, s') = \sigma_i(s)\eta_i(s')$, Value functions belong to the set $span\{\Phi^{vf}\}$ and in general, this may not hold. $\qquad \square$

### A.4. Proof of Theorem 6.1

*Theorem* 6.1. Successor Features $\psi^\pi(s, a)$ belong to an affine set and can be represented using a linear combination of basis functions and a bias.

*Proof.* Given basic state features, $\varphi : S \to \mathbb{R}^{|d|}$, the successor feature is defined as, $\psi^\pi(s, a) = \mathbb{E}_\pi[\sum_t \gamma^t \varphi(s_{t+1})]$. It can be correspondingly connected to successor measures as $\psi^\pi(s, a) = \sum_{s'} M(s, a, s') \varphi(s')$ (replace $\sum_{s'}$ with $\int_{s'}$ for continuous domains). In Linear algebra notations, let $M^\pi$ be a $(S \times A) \times S$ dimensional matrix representing successor measure. Define $\Phi_s$ as the $S \times d$ matrix containing $\varphi$ for each state concatenated row-wise. The $(S \times A) \times d$ matrix representing $\Psi^\pi$ can be given as,

$$\Psi^\pi = M^\pi \Phi_s$$
$$\implies \Psi^\pi = \sum_i \phi_i w_i^\pi \Phi_s \qquad (M^\pi \text{ is affine for basis } \phi)$$
$$\implies \Psi^\pi = \sum_{s'} \sum_i \phi_i(\cdot, \cdot, s') w_i^\pi \varphi(s')$$
$$\implies \Psi^\pi = \sum_i \sum_{s'} \phi_i(\cdot, \cdot, s') \varphi(s') w_i^\pi$$
$$\implies \Psi^\pi = \sum_i \phi_{\psi, i} w_i^\pi \qquad (\phi_\psi = \sum_{s'} \phi_i(\cdot, \cdot, s') \varphi(s'))$$
$$\implies \Psi^\pi = \Phi_\psi w^\pi$$

Hence, the successor features are affine with policy independent basis $\Phi_\psi$. $\qquad \square$

## A.5. Proof of Theorem 6.3

*Theorem* 6.3. If $M^\pi(s, a, s^+) = \phi(s, a, s^+) w^\pi$ and $\phi(s, a, s^+) = \phi_\psi(s, a)^T \phi_s(s^+)$, the successor feature $\psi^\pi(s, a) = \phi_\psi(s, a) w^\pi$ for the basic feature $\phi_s(s)^T (\phi_s \phi_s^T)^{-1}$.

*Proof.* Consider $\phi(s, a, s^+) \in \mathbb{R}^d$ as the set of $d - 1$ basis vectors and the bias with $w^\pi \in \mathbb{R}^d$ being the $d - 1$ weights to combine the basis and $w_d^\pi = 1$. Clearly from Theorem 4.2, $M^\pi(s, a, s^+)$ can be represented as $\phi(s, a, s^+) w^\pi$. Further, $\phi(s, a, s^+) = \phi_\psi(s, a)^T \phi_s(s^+)$ where $\phi_\psi(s, a) \in \mathbb{R}^{d \times d}$ and $\phi_s(s^+) \in \mathbb{R}^d$. So,

$$M^\pi(s, a, s^+) = \sum_i \sum_j \phi_\psi(s, a)_{ij} \phi_s(s^+)_j w_i^\pi$$
$$\implies M^\pi(s, a, s^+) = \sum_j \sum_i \phi_\psi(s, a)_{ij} w_i^\pi \phi_s(s^+)_j$$
$$\implies M^\pi(s, a, s^+) = \sum_j \phi_\psi(s, a)_j^T w^\pi \phi_s(s^+)_j$$
$$\implies M^\pi(s, a, s^+) = \sum_j \psi^\pi(s, a)_j \phi_s(s^+)_j \qquad (\text{Writing } \phi_\psi(s, a)^T w^\pi \text{ as } \psi^\pi(s, a))$$
$$\implies M^\pi(s, a, s^+) = \psi^\pi(s, a)^T \phi_s(s^+)$$

From Lemma 6.2, $\psi^\pi(s, a)$ is the successor feature for the basic feature $\phi_s(s)^T (\phi_s \phi_s^T)^{-1}$.

Note: In continuous settings, we can use the dataset marginal density as described in Section 5. The basic features become $\phi_s(s)^T (\mathbb{E}_\rho[\phi_s \phi_s^T])^{-1}$. $\qquad \square$

## A.6. Deriving a basis for the Toy Example

Consider the MDP shown in Figure 5. The state action visitation distribution is written as $d = (d(s_0, a_0), d(s_1, a_0), d(s_0, a_1), d(s_1, a_1))^T$. The corresponding dynamics can be written as,

$$P = \begin{array}{c} \\ s_0 \\ s_1 \end{array} \overset{\displaystyle \begin{array}{cccc} s_0, a_0 & s_1, a_0 & s_0, a_1 & s_0, a_1 \end{array}}{\left[ \begin{array}{cccc} 0 & 1 & 1 & 0 \\ 1 & 0 & 0 & 1 \end{array} \right]}$$

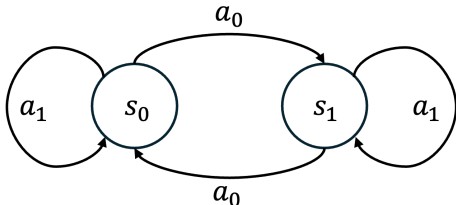

*Figure 5.* The Toy MDP described in Section 4.

The Bellman Flow equation thus becomes,

$$\sum_a d(s,a) = (1-\gamma)\mu(s) + \gamma \sum_{s',a'} P(s|s',a')d(s',a')$$

$$\implies \begin{bmatrix} 1 & 1 & 0 & 0 \\ 0 & 0 & 1 & 1 \end{bmatrix} \begin{pmatrix} d(s_0,a_0) \\ d(s_1,a_0) \\ d(s_0,a_1) \\ d(s_1,a_1) \end{pmatrix} = (1-\gamma)\begin{pmatrix} \mu(s_0) \\ \mu(s_1) \end{pmatrix} + \gamma \begin{bmatrix} 0 & 1 & 1 & 0 \\ 1 & 0 & 0 & 1 \end{bmatrix} \begin{pmatrix} d(s_0,a_0) \\ d(s_1,a_0) \\ d(s_0,a_1) \\ d(s_1,a_1) \end{pmatrix}$$

$$\implies \begin{bmatrix} 1 & 1-\gamma & -\gamma & 0 \\ -\gamma & 0 & 1 & 1-\gamma \end{bmatrix} \begin{pmatrix} d(s_0,a_0) \\ d(s_1,a_0) \\ d(s_0,a_1) \\ d(s_1,a_1) \end{pmatrix} = (1-\gamma)\begin{pmatrix} \mu(s_0) \\ \mu(s_1) \end{pmatrix}$$

This affine equation can be solved in closed form using Gauss Elimination to obtain

$$\begin{pmatrix} d(s_0,a_0) \\ d(s_1,a_0) \\ d(s_0,a_1) \\ d(s_1,a_1) \end{pmatrix} = w_1 \begin{pmatrix} \frac{-\gamma}{1+\gamma} \\ \frac{-1}{1+\gamma} \\ 1 \\ 0 \end{pmatrix} + w_2 \begin{pmatrix} \frac{-1}{1+\gamma} \\ \frac{-\gamma}{1+\gamma} \\ 0 \\ 1 \end{pmatrix} + \begin{pmatrix} \frac{\mu(s_0)+\gamma\mu(s_1)}{1+\gamma} \\ \frac{\mu(s_1)+\gamma\mu(s_0)}{1+\gamma} \\ 0 \\ 0 \end{pmatrix}. \tag{21}$$

## B. Experimental Details

### B.1. Environments

#### B.1.1. GRIDWORLDS

We use `https://github.com/facebookresearch/controllable_agent` code-base to build upon the grid-world and 4 room experiments. The task is to reach a goal state that is randomly sampled at the beginning of every episode. The reward function is 0 at all non-goal states while 1 at goal states. The episode length for these tasks are 200.

The state representation is given by $(x,y)$ which are scaled down to be in $[0,1]$. The action space consists of *five* actions: $\{up, right, down, left, stay\}$.

#### B.1.2. FETCH

We build on top of `https://github.com/ahmed-touati/controllable_agent` which contains the Fetch environments with discretized action spaces. The state space is unchanged but the action space is discretized to produce manhattan style movements i.e. move one-coordinate at a time. These six actions are mapped to the true actions of Fetch as: $\{0 : [1,0,0,0], 1 : [0,1,0,0], 2 : [0,0,1,0], 3 : [-1,0,0,0], 4 : [0,-1,0,0], 5 : [0,0,-1,0]\}$. Fetch has an episode length of 50.

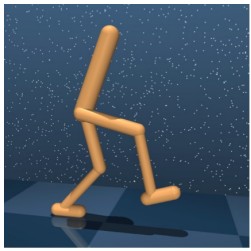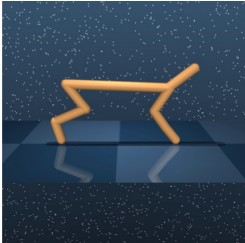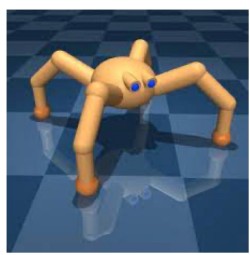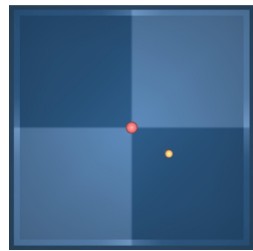

*Figure 6.* **DM Control Environments**: Visual rendering of each of the four DM Control environments we use: (from left to right) Walker, Cheetah, Quadruped, Pointmass

### B.1.3. DM-CONTROL ENVIRONMENTS

These continuous control environments have been discussed in length in DeepMind Control Suite (Tassa et al., 2018). We use these environments to provide evaluations for PSM on larger and continuous state and action spaces. The following four environments are used:

**Walker:** It has 24 dimensional state space consisting of joint positions and velocities and 6 dimensional action space where each dimension of action lies in $[-1, 1]$. The system represents a planar walker. At test time, we test the following four tasks: *Walk, Run, Stand and Flip*, each with complex dense rewards.

**Cheetah:** It has 17 dimensional state space consisting of joint positions and velocities and 6 dimensional action space where each dimension of action lies in $[-1, 1]$. The system represents a planar biped "cheetah". At test time, we test the following four tasks: *Run, Run Backward, Walk and Walk Backward*, each with complex dense rewards.

**Quadruped:** It has 78 dimensional state space consisting of joint positions and velocities and 12 dimensional action space where each dimension of action lies in $[-1, 1]$. The system represents a 3-dimensional ant with 4 legs. At test time, we test the following four tasks: *Walk, Run, Stand and Jump*, each with complex dense rewards.

**Pointmass:** The environment represents a 4-room planar grid with 4-dimensional state space $(x, y, v_x, v_y)$ and 2-dimensional action space. The four tasks that we test on are *Reach Top Left, Reach Top Right, Reach Bottom Left and Reach Bottom Right* each being goal reaching tasks for the four room centers respectively.

All DM Control tasks have an episode length of 1000.

### B.2. Datasets

**Gridworld:** The exploratory data is collected by uniformly spawning the agent and taking a random action. Each of the three method is trained on the reward-free exploratory data. At test time, a random goal is sampled and the optimal Q function is inferred by each.

**Fetch:** The exploratory data is collected by running DQN (Mnih et al., 2013) training with RND reward (Burda et al., 2019) taken from `https://github.com/iDurugkar/adversarial-intrinsic-motivation`. 20000 trajectories, each of length 50, are collected.

**DM Control:** We use publically available datasets from ExoRL Suite (Yarats et al., 2022) collected using RND exploration.

### B.3. Implementation Details

#### B.3.1. BASELINES

We consider a variety of baselines that represent different state of the art approaches for zero-shot reinforcement learning. In particular, we consider Laplacian, Forward-Backward, and HILP.

1. **Laplacian (Wu et al., 2018; Koren, 2003):** This method constructs a graph Laplacian for the MDP induced by a random policy. Eigenfunctions of this graph Laplacian gives a representation for each state $\phi(s)$, or the state feature. These state-features are used to learn the successor features; and trained to optimize a family of reward functions $r(s) = \langle \phi(s) \cdot z \rangle$, where $z$ is usually sampled from a unit hypersphere uniformly (same for all baselines). The reward functions are optimized

via TD3.

2. **Forward-Backward** ([Blier et al., 2021a](); [Touati & Ollivier, 2021a](); [Touati et al., 2023]()): Forward-backward algorithm takes a slightly different perspective: instead of training a state-representation first, a mapping is defined between reward function to a latent variable ($z = \sum_s \phi(s).r(s)$) and the optimal policy for the reward function is set to $\pi_z$, i.e the policy conditioned on the corresponding latent variable $z$. Training for optimizing all reward functions in this class allows for state-features and successor-features to coemerge. The reward functions are optimized via TD3.

3. **HILP** ([Park et al., 2024a]()): Instead of letting the state-features coemerge as in FB, HILP proposes to learn features from offline datasets that are sufficient for goal reaching. Thus, two states are close to each other if they are reachable in a few steps according to environmental dynamics. HILP uses a specialized offline RL algorithm with different discounting to learn these state features which could explain its benefit in some datasets where TD3 is not suitable for offline learning.

**Implementation:** We build upon the codebase for FB https://github.com/facebookresearch/controllable_agent and implement all the algorithms under a uniform setup for network architectures and same hyperparameters for shared modules across the algorithms. We keep the same method agnostic hyperparameters and use the author-suggested method-specfic hyperparameters. The hyperparameters for all methods can be found here:

*Table 2.* Hyperparameters for baselines and PSM.

| Hyperparameter | Value |
|---|---|
| Replay buffer size | $5 \times 10^6$ ($10 \times 10^6$ for maze) |
| Representation dimension | 128 |
| Batch size | 1024 |
| Discount factor $\gamma$ | 0.98 (0.99 for maze) |
| Optimizer | Adam |
| Learning rate | $3 \times 10^{-4}$ |
| Momentum coefficient for target networks | 0.99 |
| Stddev $\sigma$ for policy smoothing | 0.2 |
| Truncation level for policy smoothing | 0.3 |
| Number of gradient steps | $2 \times 10^6$ |
| Batch size for task inference | $10^4$ |
| Regularization weight for orthonormality loss (ensures diversity) | 1 |
| **FB specific hyperparameters** | |
| Hidden units ($F$) | 1024 |
| Number of layers ($F$) | 3 |
| Hidden units ($b$) | 256 |
| Number of layers ($b$) | 2 |
| **HILP specific hyperparameters** | |
| Hidden units ($\phi$) | 256 |
| Number of layers ($\phi$) | 2 |
| Hidden units ($\psi$) | 1024 |
| Number of layers ($\psi$) | 3 |
| Discount Factor for $\phi$ | 0.96 |
| Discount Factor for $\psi$ | 0.98 (0.99 for maze) |
| Loss type | Q-loss |
| **PSM specific hyperparameters** | |
| Hidden units ($\phi, b$) | 1024 |
| Number of layers ($\phi, b$) | 3 |
| Hidden units ($w$) | 1024 |
| Number of layers ($w$) | 3 |
| Double GD lr | 1e-4 |

**Proto Successor Measures (PSM):** PSM differs from baselines in that we learn richer representations compared to Laplacian or HILP as we are not biased by behavior policy or only learn representations sufficient for goal reaching. Compared to FB, our representation learning phase is more stable as we learn representations by Bellman evaluation backups

and do not need Bellman optimality backups. Thus, our approach is not susceptible to learning instabilities that arise from overestimation that is common in Deep RL and makes stabilizing FB hard.The hyperparameters are discussed in Appendix Table 2.

### B.3.2. PSM REPRESENTATION LEARNING PSUEDOCODE

```python
def psm_loss(
    self,
    obs: torch.Tensor,
    action: torch.Tensor,
    discount: torch.Tensor,
    next_obs: torch.Tensor,
    next_goal: torch.Tensor,
    z: torch.Tensor,
    step: int
) -> tp.Dict[str, float]:
    metrics: tp.Dict[str, float] = {}
    # Create a batch_size x batch_size for learning M^\pi(s,a,s+)
    idx = torch.arange(obs.shape[0]).to(obs.device)
    mesh = torch.stack(torch.meshgrid(idx, idx, indexing='xy')).T.reshape(-1, 2)
    m_obs = obs[mesh[:, 0]]
    m_next_obs = next_obs[mesh[:, 0]]
    m_action = action[mesh[:, 0]]
    m_next_goal = next_goal[mesh[:, 1]]
    perm = torch.randperm(obs.shape[0])

    # compute PSM loss
    with torch.no_grad():
        target_phi, target_b = self.psm_target(m_next_obs, m_next_goal)
        target_w = self.w_target(z)
        target_phi = target_phi[torch.arange(target_phi.shape[0]), next_actions.
squeeze(1)]
        target_b = target_b[torch.arange(target_b.shape[0]), next_actions.squeeze(1)]
        target_M = torch.einsum("sd, sd -> s", target_phi, target_w) + target_b

    phi, b = self.psm(m_obs, m_next_goal)
    phi = phi[torch.arange(phi.shape[0]), m_action.squeeze(1)]
    b = b[torch.arange(b.shape[0]), m_action.squeeze(1)]
    M = torch.einsum("sd, sd -> s", phi, self.w(z)) + b
    M = M.reshape(obs.shape[0], obs.shape[0])
    target_M = target_M.reshape(obs.shape[0], obs.shape[0])
    I = torch.eye(*M.size(), device=M.device)
    off_diag = ~I.bool()
    psm_offdiag: tp.Any = 0.5 * (M - discount * target_M)[off_diag].pow(2).mean()
    psm_diag: tp.Any = -((1 - discount) * (M.diag().unsqueeze(1))).mean()
    psm_loss = psm_offdiag + psm_diag

    # optimize PSM
    self.opt.zero_grad(set_to_none=True)
    self.actor_opt.zero_grad(set_to_none=True)
    psm_loss.backward()
    self.opt.step()
    self.actor_opt.step()
```

**Compute:** All our experiments were trained on Intel(R) Xeon(R) CPU E5-2620 v3 @ 2.40GHz CPUS and NVIDIA GeForce GTX TITAN GPUs. Each training run took around 10-12 hours.

## C. Additional Experiments

### C.1. Ablation on dimension of the affine space: $d$

We perform the experiments described in Section 7.3 for two of the conitnuous environments with varying dimensionality of the affine space (or corresponding successor feature in the inductive construction), $d$. Interestingly, the performance of PSM does not change much across different values of $d$ ranging from 32 to 256. This is in contrast to methods like HILP which sees significant drop in performance by modifying $d$.

| Environment | Task | $d = 32$ | $d = 50$ | $d = 128$ | $d = 256$ |
|---|---|---|---|---|---|
| Walker | Stand | $898.98 \pm 48.64$ | $942.85 \pm 19.43$ | $872.61 \pm 38.81$ | $911.25 \pm 32.86$ |
| | Run | $359.51 \pm 70.66$ | $392.76 \pm 31.29$ | $351.50 \pm 19.46$ | $372.39 \pm 41.29$ |
| | Walk | $825.66 \pm 60.14$ | $822.39 \pm 60.14$ | $891.44 \pm 46.81$ | $886.03 \pm 28.96$ |
| | Flip | $628.92 \pm 94.95$ | $521.78 \pm 29.06$ | $640.75 \pm 31.88$ | $593.78 \pm 27.14$ |
| | **Average** | 678.27 | 669.45 | 689.07 | **690.86** |
| Cheetah | Run | $298.98 \pm 95.63$ | $386.75 \pm 55.79$ | $276.41 \pm 70.23$ | $268.91 \pm 79.07$ |
| | Run Backward | $295.43 \pm 19.72$ | $260.13 \pm 24.93$ | $286.13 \pm 25.38$ | $290.89 \pm 14.36$ |
| | Walk | $942.12 \pm 84.25$ | $893.89 \pm 91.69$ | $887.02 \pm 59.87$ | $920.50 \pm 68.98$ |
| | Walk Backward | $978.64 \pm 8.74$ | $916.68 \pm 124.34$ | $980.90 \pm 2.04$ | $982.29 \pm 0.70$ |
| | **Average** | **628.79** | 615.61 | 607.61 | 615.64 |

*Table 3.* Table shows comparison (averaged over 5 seeds) between different representation sizes (or affine space dimensionality $d$) for PSM.

## C.2. Quantitative Results on Gridworld and Discrete Maze

We provide quantitative results for the experiments performed in Section 7.1.

**Quantitative Experiment Description:** For each randomly sampled goal, we obtain the inferred value function and the inferred policy using PSM and the baselines. At every state in the discrete space, we check if the policy inferred by these algorithms is optimal or not. The oracle or the optimal policy can be obtained by running the Bellman Ford al-

| Environment | Laplace | FB | FB-biased | PSM |
|---|---|---|---|---|
| Gridworld | $19.28 \pm 2.34$ | $14.53 \pm 0.68$ | $2.13 \pm 0.73$ | $\mathbf{2.05 \pm 1.20}$ |
| Discrete Maze | $38.47 \pm 7.01$ | $28.80 \pm 10.50$ | $12.9 \pm 1.63$ | $\mathbf{11.54 \pm 1.07}$ |

*Table 4.* Table shows **average error** (averaged over 3 seeds) for the predicted policy from different zero-shot RL methods with respect to the oracle optimal policy.

gorithm in the discrete gridworld or maze. We report (in Table 4) the average error (# incorrect policy predictions/Total # of states) for 10 randomly sampled goal (over 3 seeds).

The original FB method assumes that tasks are goal-conditioned, implicitly biasing the training to a smaller set of reward functions. We remove that bias in our gridworld experiments to ensure an apples-to-apples comparison. We present the results for the biased goal-conditioned sampling as "FB-biased" in Table 4.

As clearly seen, the average error for PSM is significantly less than the baselines which augments the qualitative results presented in Section 7.1.

