# OpenReview forum: "Proto Successor Measure: Representing the Behavior Space of an RL Agent"
_ICML.cc/2025/Conference — ICML 2025 poster_

### Official Review · Reviewer_PPqR · 2025-03-11

**Overall Recommendation:** 4

**Summary:**

The paper constructs a linear framework, "Proto Successor Measures", for classifying the space of Q functions, a generalization of successor features. The paper provides some theoretical results on PSM, showing how they can be learned from offline data and used for inference at test-time. Experiments show the advantage of PSM over Laplacian and Forward-Backward baselines on gridworlds and locomotion tasks.

**Claims And Evidence:**

Claim: Visitation distributions lie on a convex hull defined by affine constraints.

Evidence: Theorem 4.1, 4.2.

Claim: Better generalization properties

Evidence: Theorem 4.4, Fig 1 table 1.

Claim: The optimal policy can be produced for any reward function.

Evidence: This seems over-stated. At best, further samples are needed from the environment to infer a reward function. This could be problematic especially in sparse environments.

**Essential References Not Discussed:**

N/A

**Experimental Designs Or Analyses:**

Experiments are aligned with the claims.
- Visualization of the Q function in Fig 3 is nice, but how close is this to the optimal Q function?
- Fig 4: What are update steps? $3$ seeds is quite small -- adding more could strengthen the statistics.
- Table 3 helps understand different representation sizes (based on comments in final section)
- Table 4: Do you know how such errors scale with size of maze?
- How is there any "smoothness" between goals in the DMC tasks? They seem quite distinct, so how is there a generalization effect among tasks?

**Methods And Evaluation Criteria:**

The discrete codebook seems practically relevant and improves efficiency. The evaluation on diverse environments showcases the efficacy of the method in a variety of settings.

**Other Comments Or Suggestions:**

- Could you further clarify the use/meaning of the bias vector and how its presence departs from typical SF?
- Is there any generality lost in the linearity of the framework? What are possible downsides/alternatives/missing generalizations/adversarial reward functions to keep in mind?
- Can you provide an experiment showing the result of small datasets/dataset scaling?

**Other Strengths And Weaknesses:**

N/A (please see other comments)

**Questions For Authors:**

N/A

**Relation To Broader Scientific Literature:**

This paper nicely positions PSM as a "successor" to the successor features framework, which has proven itself in recent years. Generalizing the basis with policy-independence is a useful step forward for such zero-shot solutions and understanding the MDP geometry.

**Theoretical Claims:**

Many results stem from the linearity of the framework. Making use of the Bellman flow equation is nice.

Theorem 4.1: Its proof based on the Bellman flow equation and related arguments appears sound.

Cor 4.2: The affine set proof follows straightforwardly.

Theorem 4.4: The notation is somewhat confusing for me here. E.g. maybe there are missing brackets around $span \\{ \Phi^{vf} \\}$? Fixing typography $span \to \textrm{span}$ can also help readability. The proof seems to follow with a straightforward rearrangement of terms, but it is a bit opaque (to me): can you provide some further intuition on the terms like $\beta^\pi , k$, and their ratio? L692 is a bit confusingly worded I feel. Do you mean to say that you've shown the same $V^\pi$ is represented in $\\{ span \\{\Phi \\} r \\}$? The construction of $\Phi(s,s')$ from the indicator function seems trivial, but I might be missing something deeper here. Some further explanation would be appreciated! If I understand correctly, the proof is highlighting that only factorized $\Phi$ lead to value functions represented in the smaller span?


- Lets $\to$ Let's
- L709, Value $\to$ value

---

> ### Author Rebuttal · Authors · 2025-04-01
>
> We thank the reviewer for providing detailed feedback on our paper. We would like to address the concerns raised by the reviewer:
>
> > At best, further samples are needed from the environment to infer a reward function. This could be problematic especially in sparse environments. :
>
> We agree with the reviewer’s observation that practically sparse reward environments can be problematic. Mathematically, our method can be used to obtain the near-optimal policy for any reward function. But in practice, the quantity $\sum_{s’} \phi(s, a, s’)r(s’)$ is approximated by $\mathbb{E}_{s’ \sim \rho}[\phi(s, a, s’) r(s’)]$. This restricts the knowledge of reward function to only samples from the function, which could affect the optimality in some situations. We will soften the claim in the camera-ready version.
>
> >Theorem 4.4: The notation is somewhat confusing for me here. Fixing typography  can also help readability.:
>
> We thank the reviewer for pointing out that the Theorem statement might be confusing. We shall rewrite the theorem statement in the camera ready version. Through this theorem, in line with reviewer’s understanding we wished to portray that spanning successor measures using a basis covers a larger set of RL solutions than spanning value functions using a basis.
>
> > The proof seems to follow with a straightforward rearrangement of terms, but it is a bit opaque (to me):
>
> The proof establishes that for every instance of a $V^\pi \in span{\phi^{vf}}$, there exists an instance of $V^\pi$ in $\span{\Phi}r$ for some reward. While the reverse is not always true. $\beta^\pi$ is used to denote the weights of the linear combination of \phi^{vf} while representing $V^\pi$ and $k_i = \sum_{s’}r(s’)$ is used for algebraic manipulation. We appreciate the reviewer’s concern and will make the proof clearer in the camera-ready version by adding explanations for the different algebraic manipulations used.
>
>
> > Fig 4: What are update steps?  seeds is quite small -- adding more could strengthen the statistics.:
>
> Update steps means the number of learning updates made to the bases and bias. $\phi$ and $b$ were evaluated on their zero-shot performance after every fixed number of training steps.
>
> We appreciate the reviewer for pointing this out and will add more seeds during the camera ready version.
>
> > Table 4: Do you know how such errors scale with size of maze?:
>
> There are 50 and 68 cells respectively in the grid world and four room maze. So the % of wrong predictions for PSM are 4.1% and 16.97%. This is in comparison to 29.06% and 42.35% for FB and 38.56% and 56.57% for Laplace.
>
> > How is there any "smoothness" between goals in the DMC tasks? They seem quite distinct, so how is there a generalization effect among tasks?
>
> We are not sure if we understand the reviewer’s question - Most of the tasks we evaluate for DMC use dense rewards as opposed to goals. These tasks are the standard ones that prior methods (Touati et. al., 2023; Park et. al. 2024) used for evaluating unsupervised RL performance. PSM uses the unlabelled transition dataset to learn all possible behaviors (stitched trajectories). Near optimal policies for each of these tasks correspond to a particular stitching.
>
> > Could you further clarify the use/meaning of the bias vector and how its presence departs from typical SF?:
>
> The bias vector comes from the fact that the Bellman equation is affine and not linear. The significance being that a zero visitation/successor measure is not permitted which might be permitted in SF. Note that it is possible to recover the bias by representing successor measures as $\phi w$ (if the last coordinate of $w$ being learnt to be 1). But, the bias term forces a mathematically correct representation. What truly departs from SF is the absence of the assumption of the linear mapping between rewards and policy embeddings.
>
> > Is there any generality lost in the linearity of the framework? What are possible downsides/alternatives/missing generalizations/adversarial reward functions to keep in mind?:
>
> The PSM objective ideally will represent all possible successor measures contrary to SF-methods which focus on learning successor measures optimal for a particular class of reward functions (linear-in-state-representation). This means they will fail to capture the successor measure if it was not optimal for a certain reward function belonging to the class. Second, assuming linearity between rewards and zero-shot policy embeddings, fundamentally disallowing many-to-many mapping between policy and rewards.
>
>
> > Can you provide an experiment showing the result of small datasets/dataset scaling? :
>
> We run PSM on a reduced dataset (by selecting only 10%) of the trajectories in the RND dataset for walker and cheetah. While the performance on walker remains 656.08 (full dataset performance: 689.07), the performance on cheetah dips to 211.53 (full dataset performance: 626.01). We will add detailed experiments in the camera ready version.

---

### Official Review · Reviewer_UwHe · 2025-03-14

**Overall Recommendation:** 3

**Summary:**

The paper introduces Proto Successor Measure (PSM), a basis set to represent all possible behaviors of an RL agent in an environment. The key insight is that any valid successor measure must satisfy the Bellman flow equation. By rearranging the Bellman flow equation one gets an affine equation. Any solution to this affine equation can be represented as an affine combination of a basis set, where the bases and the bias term are independent of the policy. Hence, by learning this basis set, one can represent the successor measure of any policy by finding the corresponding set of linear weights $w$. Given a downstream reward, one can solve a linear program to obtain the $w$ corresponding to the successor measure for the optimal behavior, obtain the optimal Q function, and then back out the optimal policy. With these theoretical foundations, the paper introduces a practical algorithm to learn the basis functions using reward-free interaction data. This involves using a seeded RNG to draw samples from the policy space. The method is evaluated on a selection of manipulation and locomotion zero-shot RL problems from FetchReach and ExoRL, where PSM exhibits superior transfer behavior compared to prior zero-shot RL baselines.

**Claims And Evidence:**

The claims made in the submission are supported by clear and convincing evidence. Empirically, the paper claims that PSM leads to more desirable value functions and better transfer performance. The visualizations of value functions in the Gridworld and Four-room environments (Figure 3) show that PSM value functions are more concentrated than the FB and Laplace value functions. On FetchReach and ExoRL (Figure 4, Table 1), PSM achieves higher episode returns than baselines when transferring to downstream rewards. For theoretical claims, please refer to the discussion in the "Theoretical Claims" section.

**Essential References Not Discussed:**

The paper includes sufficient context for the reader to understand its contributions.

**Experimental Designs Or Analyses:**

I checked the soundness of the experimental designs, including the Gridworld analysis, the FetchReach environment, and the ExoRL benchmark.

**Methods And Evaluation Criteria:**

This paper applies PSM to the problem of zero-shot reinforcement learning, which aims to learn a representation of policies from a reward-free dataset and quickly obtain the optimal policy for some test-time reward functions without additional environment interaction. The evaluation domains properly represent this problem, where the methods are trained on some large reward-free datasets and transferred to multiple different reward functions.

**Other Comments Or Suggestions:**

1. Line 43 typo "familia r"
2. Line 89 typo "ge neralization"
3. Line 177 "Successor measures are more general than state-action visitation distributions as they encode the visitation of the policy conditioned on a starting state-action pair." This statement is inaccurate because successor measures are a special case of state-action visitation distributions (equation 2).
4. Line 189 "we do not need the first constraint on the linear program (in Equation 3) anymore" -> the second constraint.

**Other Strengths And Weaknesses:**

**Strengths**
1. This work proposes a principled approach to zero-shot RL, a problem of great interest to the community.
2. The derivations are driven by first principles and mathematically sound. The practical implementation achieves superior empirical performance compared to relevant baselines.
3. This work improves upon prior zero-shot RL methods by learning representations truly independent of the policy and in principle capable of representing all behaviors in an RL environment.

**Weaknesses**
1. The inference for the linear weights involves solving a linear program with constraints. This is harder to solve than linear regression in prior works.
2. PSM does not directly produce policies. It produces the successor measure corresponding to the optimal policy for the downstream reward. To back out a policy for a continuous domain, one needs to recover the Q value and then perform several iterations of policy optimization to take actions that maximize the Q values. In some prior work, the definition of zero-shot RL is no additional policy optimization. PSM would violate this strict definition of zero-shot RL.
3. The paper does not address the data assumption. In principle, the dataset needs to have full coverage of the state and action spaces to learn a basis set that spans the entire space of successor measures.

**Questions For Authors:**

1. How do you solve (4) in continuous space? Do you sample (s, a) from a dataset?
2. How do you determine the dimensionality of the basis set? Can you provide an ablation of the dimensionality of the bases?
3. In hyperparameters (Table 2), why does w have 3 layers? Is it represented by a neural network?

## Updates After Rebuttal
The authors have adequately addressed my questions. I will maintain my score.

**Relation To Broader Scientific Literature:**

The key contributions are built on a line of prior works on zero-shot reinforcement learning. Particularly relevant is the FB representation [1,2], which proposes to decompose a successor measure into a product of a forward representation and a backward representation. Unlike FB where the representations are conditioned on some policy representation $z$, PSM learns policy-independent basis functions, which is more desirable.

[1] Ahmed Touati, Yann Ollivier. Learning One Representation to Optimize All Rewards. NeurIPS 2021.
[2] Ahmed Touati, Jérémy Rapin, Yann Ollivier. Does Zero-Shot Reinforcement Learning Exist? ICLR 2023.

**Theoretical Claims:**

I checked the correctness of the proofs for the theoretical claims. The paper provide three sets of theoretical claims:
1.  (Theorem 4.1, Corollary 4.2, Corollary 4.3) Deriving proto successor measure as the basis set for solutions to an affine equation.
2. (Theorem 4.2) The basis value functions represent a smaller space of value functions than proto successor measures.
3. (Theorem 6.1, Lemma 6.2, Theorem 6.3) Connects PSM to successor features showing that one can decompose PSM into successor features corresponding to some state features. These proofs largely follow the mechanisms in [1].

The proofs for these claims appear to be sound.

[1] Ahmed Touati, Jérémy Rapin, Yann Ollivier. Does Zero-Shot Reinforcement Learning Exist? ICLR 2023.

---

> ### Author Rebuttal · Authors · 2025-04-01
>
> We appreciate the reviewer for providing detailed feedback on our paper. We would like to clarify all the questions raised by the reviewer:
>
> > The inference for the linear weights involves solving a linear program with constraints. This is harder to solve than linear regression in prior works. :
>
> Yes, while we agree that the full PSM objective requires solving a constraint LP, which can be harder than simple linear regression as in some of the prior work, other methods relying on linear regression rely on the assumption that policy embeddings z have a linear relation to reward functions. Enforcing this assumption in PSM, as we have shown in Theorem 6.3 (Line 318-328 right column), PSM can also enable inference through linear regression.
>
>
> > PSM does not directly produce policies. It produces the successor measure corresponding to the optimal policy for the downstream reward. To back out a policy for a continuous domain, one needs to recover the Q value and then perform several iterations of policy optimization to take actions that maximize the Q values. In some prior work, the definition of zero-shot RL is no additional policy optimization. PSM would violate this strict definition of zero-shot RL. :
>
> Following our explanation in comment 1 above, the full objective of PSM does require additional optimization to obtain policy from successor measures, but these optimizations are much easier than solving an RL problem and similar to a policy evaluation problem. With the additional assumption of rewards linear in state features obtained by PSM, we can enable true zero-shot inference without additional optimization. The key benefit of PSM lies in its flexibility to accurately represent all behaviors of an agent and with an additional assumption can also outperform prior works in zero-shot RL.
>
>
> > The paper does not address the data assumption. In principle, the dataset needs to have full coverage of the state and action spaces to learn a basis set that spans the entire space of successor measures:
>
> The capabilities of any unsupervised learning algorithm is intrinsically tied to the dataset it is learning from. PSM and its unsupervised RL counterparts FB and HILP all are limited by dataset but PSM aims to learn to represent a more diverse range of skills as a virtue of its objective. But this very property can be beneficial in practice: When using real world dataset to learn skills we would avoid focusing on skills that are irrelevant in the real world.  With complete coverage of the space, PSM will learn the basis that attempts to span the entire space of successor measures.
>
> > Typos:
>
> We thank the reviewer for pointing out these typos. We shall fix them in the camera-ready version.
>
> > How do you solve (4) in continuous space? Do you sample (s, a) from a dataset?
>
> In both continuous and discrete spaces, we sample (s, a) from the dataset.
>
> > How do you determine the dimensionality of the basis set? Can you provide an ablation of the dimensionality of the bases?
>
> We have provided an ablation for the dimensionality of the bases in Appendix C.1.  The dimensionality of the bases can potentially vary depending on the dynamics. For instance, If the dynamics is very regular (symmetric and Lipschitz as a lot of the real world domains are), the dimensionality may be way less than the mathematical limit of S x A. An example can be constructed similar to the one in Section B.2 in [1].
>
> [1]: Touati, Ahmed, and Yann Ollivier. "Learning one representation to optimize all rewards." Advances in Neural Information Processing Systems 34 (2021): 13-23.
>
> > In hyperparameters (Table 2), why does w have 3 layers? Is it represented by a neural network?
>
> During pretraining i.e., obtaining the bases of PSM, we represent w as a function of the policy representation: the discrete code c. As a result, we represent w as a neural network which is a mapping from c to $\mathbb{R}^d$.

---

### Official Review · Reviewer_Gftv · 2025-03-14

**Overall Recommendation:** 3

**Summary:**

The paper investigates representation learning in RL with the aim of performing zero-shot learning: computing optimal policies on downstream tasks wihout any further training. Building on earlier works about representation learning in a reward-free setting, especially on successor representations, it proposes a new representation called "proto-successor measure" (PSM) and gives a procedure to learn this representation. It is based on the observation that successor measures (the discounted distribution of states under a given policy) satisfy a Bellman equation which is independent of the policy. As solutions of linear systems, successor measures and related objects thus span an affine subspace. The proto-successor measure consists then in computing an approximation of this affine subspace. This theory is completed with experimental works in gridworld and continuous control environments, exhibiting better performance than forward-backward, HILP and Laplacian representations, which are popular representations that attracted a lot of attention recently.

## Update after rebutal:
I think the authors for addressing my last questions. I agree with the authors that PSM may bring a complementary and novel idea to the FB representation of Touati \& al, so I will maintain my score (weak accept).

**Claims And Evidence:**

The authors claim their work on the PSM representation gives a "novel, mathematically complete perspective" on representation learning in RL, together with an efficient algorithm. While I think the present work has an interest, I am a bit more reserved about its ground-breaking aspect.

From a theoretical perspective, I think the main contribution is in the observation that there exists a Bellman equation independent of the policy. This is a simple fact which could nonetheless have powerful consequences and to the best of my knowledge this is indeed new. However I don't think the rest of the theoretical content goes much deeper than this: the main result Theorem 4.1 and its corollaries for instance are simple statements that solutions of a linear system span an affine subpsace. So I think the "mathematically complete" mention is a bit exaggerated here. The paper would require a much more quantitative analysis to deserve that mention.


When it comes to the experiments, the authors underline the apparently better performances of PSM over other representations like the forward-backward (FB) model of Touati \& al, but I am not sure this comparison is really fair. I believe the generalization power of PSM is a bit different: as the authors write in Theorem 6.3, the PSM representation factorizes the successor measure as $M^{\pi}(s,a,s^{+}) = \phi(s,a,s^{+}) w^{\pi}$, while the FB model adds a further factorization in that $M^{\pi}(s,a,s^{+}) = \psi^{\pi}(s,a) \phi(s^{+})$ (I consider a tabular setting). Therefore I think PSM allows a generalization over policies but it remains intrinsically a high-dimensional ($S^2 A$), more complex representation than FB which generalizes both over policies and state-actions. Thus it hardly comes to as a surprise that PSM provides more expressive representations than all other benchmarks considered here, and I suspect that in a tabular a setting the sample-complexity of learning PSM could scale as $\Theta(S^2 A)$.

**Essential References Not Discussed:**

I think the paper already discusses the most important references in relation to the subject.

**Experimental Designs Or Analyses:**

I don't have more to say than in "Methods and Evaluation Criteria".

**Methods And Evaluation Criteria:**

The overall method to learn the PSM representation makes sense. I believe however there is a typo in the loss function p.5, I think there is a square as well as a $(1-\gamma)$ factor missing. Also I haven't quite understood the idea of the discrete codebook of policies, and in particular why the "approach provably samples from among all possible deterministic policies".

For evaluation criteria I don't have concerns other than the one above: I am not sure think PSM really compares with FB, Laplacian and the likes.

**Other Comments Or Suggestions:**

Overall I think the basic idea of this paper, that successor measures span an affine subspace, has a good potential and the experiments are promising. However I think the theoretical content is quite simple so I would suggest putting the focus on the experimental work. For instance, I don't think Section 3.2 on affine spaces rally brings a lot to the paper. I would assume most researchers in machine learning know what an affine space is.

Secondly, the paper puts the emphasis on state-action distributions and regards the successor measure as a secondary. I believe this should be the other way around, the successor measure is the central object and state-action distributions a byproduct of it.

Finally I have noticed a few typos here and there: "familia r' line 043, "ideplogy" p.8

**Other Strengths And Weaknesses:**

The main idea of the paper is simple but interesting. The paper is in general pretty clear.

**Questions For Authors:**

1. I'd like primarily the authors to address the concern raised above: is it true as I think that PSM and FB do not play in the same league exactly, since the features of PSM are functions of $(s,a,s')$?  Is it really possible to learn PSM as efficiently as FB?

**Relation To Broader Scientific Literature:**

The paper seems to be heavily inspired by works on Laplacian representations as well as the recent line of work of Touati \& al who proposed the forward-backward model representation to perform zero-shot learning. The present paper starts by considering the same object, the successor measure, which for a fixed policy is the operator that maps a reward to its value function. It has appeared quite often in the literature. The forward-backward model of Touati \& al. aims to compactly represent the successor measure of all optimal policies though what is essentially a low-rank matrix in the tabular setting. The PSM representation bears resemblance in that it attempts to represent successor measures for different policies, but the representation is now low-rank. It leverages the fact that successor measures span an affine subpsace, which I think has not been observed in the work of Touati \& al.

**Theoretical Claims:**

I haven't checked all the proofs but I think the results are simple enough to be correct.

---

> ### Author Rebuttal · Authors · 2025-04-01
>
> We thank the reviewer for taking time to review our paper and providing detailed feedback. We would like to address the concerns of the reviewer below:
>
> > I think the "mathematically complete" mention is a bit exaggerated here. The paper would require a much more quantitative analysis to deserve that mention.:
>
> By “mathematically complete,” we mean that the representation learning framework is derived without making any assumptions on (1) dynamics, (2) reward functions, or (3) the complexity of the state or action space. Since the practical algorithm has some approximations to make the framework more tractable, starting with the low-rank approximation of the bases space itself, we thank the reviewer for pointing this out and understand it might be construed to mean something stronger.Wee shall soften and clarify this language in the camera-ready version.
>
> > I am not sure think PSM really compares with FB, Laplacian and the likes:
>
> We respectfully disagree with the reviewer that PSM does not really compare with FB, Laplacian. In all our experiments, the respective functions, ($\phi, b$ for PSM or $F, B$ for FB) are represented using neural networks with a similar number of parameters, thus working with similar hypothesis spaces. While PSM requires training the bases with networks taking in (s, a, s’), FB requires training two networks, one with input (s, a, z) and the other with (s). The dimension of z is similar to (often more than) s  (128 in our experiments). This observation, in some ways, makes FB require more parameters than PSM. But, in any case, FB and other SF based approaches like Laplace and HILP seem to be the most relevant prior work in terms of zero-shot RL that PSM can compare against.
>
> > Typos:
>
> We thank the reviewer for pointing out the typos. We shall correct them in the camera-ready version.
>
> > I think the theoretical content is quite simple so I would suggest putting the focus on the experimental work.:
>
> We are glad to see the reviewer likes our observation that successor measures  span an affine space. While the idea may seem simple in hindsight, it has (to the best of our knowledge) remained unexplored in the RL literature. Hence we decided to spend more time building towards insights we can obtain from the affine-space idea of successor measures - for instance we are able to show that with the same basis dimensions, we can span a larger space of values than methods which learn a basis space across value functions.
>
>
> > The paper puts the emphasis on state-action distributions and regards the successor measure as a secondary. I believe this should be the other way around, the successor measure is the central object and state-action distributions a byproduct of it.:
>
> We are aligned with the reviewer's observation that successor measures are the main object we end up learning to represent. For unsupervised learning, it is preferable to learn to represent successor measures as they capture more information that visitation distributions. Our choice of starting with visitation distributions was motivated by the large number of works that explore the idea of RL as a linear program and goes to show the same idea can be used for unsupervised representation learning. The familiarity of readers with that literature motivated us to start from visitation distribution to facilitate understanding. We will clarify the writing to place more emphasis on the successor measure.
>
>
> > Is it true as I think that PSM and FB do not play in the same league exactly, since the features of PSM are functions of (s, a, s’)? Is it really possible to learn PSM as efficiently as FB?:
>
> As discussed earlier, FB and PSM are parameterized with neural networks with a similar number of parameters, in fact, depending on the sizes of s and z, FB may be parameterized with a larger network. Empirically it has been observed that FB is difficult to stabilize as it is optimizing for a moving reward function ($B^{-1} z$) while PSM does not face this issue as it does not tie the policy latents to reward functions. Moreover, FB uses the policy optimized for the corresponding reward (through Q function maximization) to sample actions during its training. This means that FB is susceptible to overestimation issues occurring due to Q function maximization.

---

> > ### Comment · Reviewer_Gftv · 2025-04-04
> >
> > I thank the authors for their response. My concern about the comparison of PSM and FB has been answered, but I have still a few comments. The authors did not address my remark regarding the discrete codebook.
> >
> > About FB and PSM:
> > I understand that on the experiments, PSM shows better performance with similar number of parameters, but these remain specific examples. I don't take this as a proof that PSM will in general always perform better that FB. When saying that I am not sure  FB compares with PSM, I did not mean it as a critic, but more as the suggestion it could be a complementary idea. I wonder for instance if the fact that all successor measures satisfy the same affine equation could shed new light on FB and Laplacian representations.
> >
> > About the discrete codebook: I probably could have been clearer in my review. I understand the idea of the codebook but I wasn't sure to understand Eq. (8), which I interpreted literally. I guess the authors mean that for each state the action are sampled uniformly at random with a seed depending on z and s?
> >
> > With hindsight and from the other reviewer remarks, I think the discrete codebook idea is a key feature of the paper and perhaps the biggest difference with FB. There should be a discussion about the size, as suggested by reviewer MScc. The sampling of policies is also what makes me think why FB could still be preferable in some cases, since it directly attempts to find optimal policies. On the other hand, why the codebook of sampled policies would give a meaningful representation, if the codebook is too small?

---

> > > ### Author Response · Authors · 2025-04-07
> > >
> > > We thank the reviewer for their response. We address all the concerns raised by the reviewer  as follows:
> > >
> > > > The authors did not address my remark regarding the discrete codebook.
> > >
> > > We apologize for missing your remark about the discrete codebook. The discrete codebook was a neat technique to achieve a sampling distribution of all possible deterministic policies. It is implemented by sampling an integer $z$, constructing a seed using this integer and the hash of the state, and using this seed to sample from a uniform distribution. If $z \in \mathbb{Z}^+$, or $z$ covers all seeds, we get all possible deterministic policies. In practice, $z \leq 2^{16}$ has worked well for all the domains.
> > >
> > > > I don't take this as a proof that PSM will in general always perform better that FB.
> > >
> > > The reviewer is correct that PSM performing better than FB empirically does not prove the PSM will in general always perform better than FB. In our experiments across multiple domains (both discrete and continuous), we found PSM to consistently outperform FB while also requiring much less hyperparameter tuning. Theoretically, we hypothesize the following reasons why PSM is better than FB:
> > >
> > > (1) FB inherits limitations from SF-based approaches due to a similar linear mapping between rewards and policy embeddings. This mapping is incorrect as the rewards to policy mapping is many to one rather than one to one.
> > >
> > > (2) During training, FB samples from policies that are optimized (through Q function maximization) for the corresponding reward ($B^{-1} z$). This maximization step is prone to overestimation. PSM on the other hand, uses a milder form of off-policy evaluation that is more stable.
> > >
> > > (3) FB optimizes its actor for a changing reward function ($B^{-1} z$) as B gets updated every iteration.
> > >
> > > We will reassure our claim to say that PSM is competitive alternative; as there a number of uninterpretable factors for deep representation learning that might make one method over the others.
> > >
> > > > I wasn't sure to understand Eq. (8), which I interpreted literally. I guess the authors mean that for each state the action are sampled uniformly at random with a seed depending on z and s
> > >
> > > Yes, the reviewer is correct. We specify the seed using the state $s$ and the integer code $z$ to approximate the distribution over all policies.
> > >
> > > >  Discrete codebook is perhaps the biggest difference with FB
> > >
> > > While a discrete codebook in the practical implementation of PSM is a new idea than FB, we believe it is only one of the possible ways to implement PSM. PSM is a more general framework that leverages the structure of visitation and successor measure in representation learning. Our previous answer also points out some reasons that PSM is potentially a stronger approach than FB, theoretically.
> > >
> > > > The sampling of policies is also what makes me think why FB could still be preferable in some cases, since it directly attempts to find optimal policies. On the other hand, why the codebook of sampled policies would give a meaningful representation, if the codebook is too small?
> > >
> > > In practice, FB requires a number of practical implementation tricks to make learning stable and performant. As an example to make test-time policies performant, FB samples from a prior over reward functions that consists of 50% rewards that are goal-reaching. Besides that, FB also requires careful architecture design such as using a layernorm in the first layer. These are by-products of some of the theoretical reasons we list in our first comment.
> > >
> > > PSM is practically stable to train and does not require bellman optimality backups, which are prone to overestimation in the offline setting. We believe it's better to think of PSM and FB as complementary ways, and future work can build on both of these ideas together. For the codebook, we use a codebook of $2^{16}$ policies which we found to be sufficient for good representations and the size of the codebook has minimal effect on computational requirements for training.

---

### Official Review · Reviewer_MScc · 2025-03-23

**Overall Recommendation:** 3

**Summary:**

This paper studies the reward-free RL problem and proposes the concept of proto successor measure (PSM), which is built on the idea of proto value functions in (Mahadevan and Maggioni, 2007) and serves as the basis set of all the possible visitation distributions in a given RL environment. By learning this basis set, given a reward function at the inference stage, solving for an optimal policy is equivalent to solving a constrained linear program, where both the objective function and the Bellman flow constraint can be expressed in terms of this basis. Accordingly, this paper proposes to implement the PSM by integrating two components: (1) Using the measure loss similar to that in (Touati & Ollivier, 2021), one can learn the weight that corresponds to a given policy under some learned PSM. (2) To learn the basis set in PSM via pre-training, this paper proposes to construct a discrete codebook of policies, which serves as a way to simulate uniform sampling over policies.

The proposed PSM is evaluated on both discrete and continuous control environments, including gridworld-like problems, Fetch-Reach tasks, and the ExoRL suite. Through experiments, the PSM is shown to be comparable or outperform the recent benchmark reward-free RL methods.

## update after rebuttal

I appreciate the authors for the additional response and for answering my follow-up questions. The additional sanity check of FB and the clarification on linear mapping are helpful.

On the other hand, I am still not convinced by the explanation about overestimation bias from the off-policy learning perspective (FB in (Touati et al., 2023) was implemented based on TD3, and there are existing techniques to largely avoid overestimation) and the argument “...This instability is absent in PSM as PSM performs off-policy evaluation using samples from the codebook policy…”

After going through all the reviews and the rebuttal responses again, overall, the idea of PSM appears to be a useful addition to the reward-free RL literature with good empirical support. Hence, I have increased my score accordingly and lean towards acceptance.

**Claims And Evidence:**

The main theoretical claims of this paper (Theorems 4.1 & 4.4, Corollaries 4.2-4.3, and Theorems 6.1 & 6.3) are supported by the proofs in the appendix.

**Essential References Not Discussed:**

As far as I know, most of the recent relevant works on learning reward-free RL are cited in this paper. That being said, to better describe the contributions of this papers, there are two things to clarify:

- Comparison with successor feature methods: While Section 6 points out that SF can be viewed as a special case of PSM, it is not well explained why it is beneficial to consider PSM in general. Theoretically speaking, using SF, like in (Lehnert & Littman, 2020; Hoang et al., 2021; Alegre et al., 2022; Reinke & Alameda-Pineda, 2021), can already be sufficient to achieve similar zero-shot generalization.
- Comparison with FB representation: While I can appreciate the idea of PSM, it remains unclear to me why PSM is a better approach than the FB representation. Indeed, as also shown in Table 1, the two approaches are just comparable in most of the tasks. Moreover, compared to FB, PSM requires solving an additional constrained linear program to get a policy at inference time. Accordingly, it appears that FB could be preferred in practice. While it is mentioned in Lines 155-163 that “FB ties the reward with the representation of the optimal policy derived using Q function maximization which can lead to overestimation issues and instability during training as a result of Bellman optimality backups,” I do not find this argument convincing. It would be helpful to provide more evidence to justify this argument.

**Experimental Designs Or Analyses:**

I have checked the experimental designs in Section 7. Most of them look reasonable.

However, there are two things that require further explanation:
- Representative reward functions at test time: As mentioned above, one concern is on whether the small set of reward functions considered in Section 7 for each environment is indeed representative enough. Specifically, like the existing reward-free RL literature, this paper evaluates the capability of zero-shot generalization by testing the performance of PSM and other baselines across a few difference reward functions (e.g., for the Walker environment, take the reward functions of the four underlying tasks Stand, Run, Walk, and Flip). While I know this evaluation procedure is also adopted by some prior works (e.g., (Touati & Ollivier, 2021) and (Touati et al., 2023)), to make a more thorough comparison, it would be good to compare a more diverse set of reward functions for each environment.

- Comparison with successor feature methods: The PSM approach is highly related to the general method of Successor Features (SF). Therefore, it would be helpful to compare PSM with SF-based approaches, e.g., (Alegre et al., 2022) to demonstrate the benefits of learning this basis set.

- Regarding the qualitative results in Figure 3, it appears that FB cannot generalize well in the four-room environment. However, this is quite different from the results provided in the original FB paper (Touati & Ollivier, 2021). It is unclear where the differences come from. It would be good to describe this in more detail to ensure a fair comparison.



(Alegre et al., 2022) Lucas N. Alegre, Ana L. C. Bazzan, and Bruno C. da Silva, “Optimistic Linear Support and Successor Features as a Basis for Optimal Policy Transfer,” ICML 2022.

**Methods And Evaluation Criteria:**

- Like the existing reward-free RL literature, this paper evaluates the capability of zero-shot generalization by testing the performance of PSM and other baselines across a few difference reward functions (e.g., for the Walker environment, take the reward functions of the four underlying tasks Stand, Run, Walk, and Flip). One concern is that it is unclear whether this small set of reward functions is already representative enough. While I know this evaluation procedure is also adopted by some prior works (e.g., (Touati & Ollivier, 2021) and (Touati et al., 2023)), to make a more thorough comparison, it would be good to compare a more diverse set of reward functions for each environment.

- All the pre-training is done on a shared offline dataset constructed using a fixed behavior policy, and this looks fair to all the algorithms.

**Other Comments Or Suggestions:**

- The statement of Theorem 4.4 is quite hard to parse,  and the presentation of it can be improved. This is partially because the notations are quite confusing.
- Equation (3) can be reorganized. There are two separate “such that” which can be merged into one.
- An ablation study on the need for the discrete policy codebook would be helpful.

**Other Strengths And Weaknesses:**

All the strengths and weaknesses are provided in the sections above.

**Questions For Authors:**

Please see the above questions on the comparison with SF-based and FB-based methods, the baselines for evaluation, as well as the experimental configuration for the test-time evaluation. I will be willing to reconsider my evaluation if the authors can address these issues.

**Relation To Broader Scientific Literature:**

The main contribution of this paper is to offer another perspective of solving reward-free RL by pre-training a basis set called Proto Successor Measure. While this line of research can be traced back to the classic proto value functions (Mahadevan and Maggioni, 2007), this paper offers a practical implementation that scales the idea of proto value function to a broader class of RL problems, including high-dimensional robot control. Given the recent attention on RL foundation models, I find this paper to be quite nice in contributing to the general literature of RL foundation models.

**Theoretical Claims:**

- The main theoretical results of the paper are two-fold: (1) The set of possible visitation distributions form an affine set (Section 4); (2) The classic successor features can also be represented using a similar basis set.

- I have checked the proofs in the appendix up to Appendix A.3, and I did not spot any major issues regarding the correctness of the analysis.

---

> ### Author Rebuttal · Authors · 2025-04-01
>
> We thank the reviewer for their detailed feedback. We would like to address all the concerns of the reviewer below:
>
> > It would be good to compare a more diverse set of reward functions for each environment.:
>
> We appreciate the reviewer’s suggestion and agree that our original evaluation using four tasks per environment, though consistent with prior works, could be further expanded for stronger claims. To directly address this concern, **we have conducted additional experiments on a significantly larger and more diverse set of reward functions (10+ tasks) in the Walker environment**. Our extended results confirm that PSM consistently outperforms or matches the baselines. The mean performance  on these 10+ tasks are (across 5 seeds), PSM: 523 +- 48.86, FB: 516.95 +- 45.61 and Laplace: 250.43 +- 30.58. We shall include detailed results with diverse tasks for each of the domains in the camera ready.
>
>
> > Comparison with successor feature methods:
>
> We would like to point out that the baselines that we considered are in fact SF-based approaches. “Laplace” uses Laplace eigenvectors as state features while HILP has its own objective for learning state features. Both of them learn Successor Features for these corresponding state features. Moreover, FB is also built off an SF-based approach. We acknowledge the reviewer and will make it clearer in the paper. Additionally, **we are adding experiments on a couple of other SF-based approaches (best performant as per (Touati et. al., 2023)**: one that uses one-step dynamics predictability (method named “Trans”) and one that uses SVD decomposition of successor representation (method named “SVD”). The mean performance for these baselines across all DMC tasks are (across 5 seeds): Trans: 522.10 and SVD: 524.96. PSM outperforms them by a significant margin with a mean performance: 607.60. We will add the detailed results in the camera ready.
>
> > However, this is quite different from the results provided in the original FB paper. It is unclear where the differences come from. :
>
> We thank the reviewer for noting this difference. We would like to clarify that the original FB paper does not report quantitative performance metrics comprehensively on the four-room environment but rather highlights qualitative successes with select examples.
>
>
> Additionally, to ensure fair comparison with FB, we deliberately modified its training strategy. The original FB method assumes that tasks are goal-conditioned, implicitly biasing the training to a smaller set of reward functions. Since PSM makes no such assumption, we adapted FB training to uniformly sample reward functions to ensure an apples-to-apples comparison. This explains the observed discrepancy in FB performance. We will clearly document these modifications in the final version. We shall include the results for the biased sampling in FB in the final version.
>
> > While Section 6 points out that SF can be viewed as a special case of PSM, it is not well explained why it is beneficial to consider PSM in general. :
>
> We appreciate the reviewer highlighting the need for clearer explanation of the advantages of PSM over traditional SF approaches. SF-based methods rely on two restrictive assumptions:
>
>
> (1) Sufficient Feature Representation: SF assumes either predefined state features or features learned through an auxiliary objective, presuming these features adequately span all possible reward functions which  limits representational expressiveness.
>
>
> (2) Linear Mapping between Rewards and Policies: SF-based methods condition policies linearly on feature weights, assuming a one-to-one mapping between optimal policies and rewards. Practically, the mapping is many-to-many; multiple rewards can share the same optimal policy and vice versa, making such linearity limiting.
>
>
> In contrast, PSM avoids both assumptions and directly aiming to accurately capture all feasible visitation distributions.
>
> > It remains unclear to me why PSM is a better approach than the FB representation:
>
> (1) FB inherits limitations from SF-based approaches described above due to a similar linear mapping. (2) During training, FB samples from policies that are optimized (through Q function maximization) for the corresponding reward ($B^{-1} z$). This maximization step is prone to overestimation (3) FB is unstable due to optimizes for a changing reward function as B gets updated.
>
> > The statement of Theorem 4.4 is quite hard to parse, and the presentation of it can be improved. This is partially because the notations are quite confusing. :
>
> We thank the reviewer for pointing this out. We agree and will revise the statement and the notation more clearly in Theorem 4.4 in the final version to improve clarity.
>
> > An ablation study on the need for the discrete policy codebook would be helpful.:
>
> The discrete codebook is an essential component of our method that makes the PSM training tractable. We will add an ablation on the size of the codebook in the final version.

---

> > ### Comment · Reviewer_MScc · 2025-04-04
> >
> > Thanks to the authors for the detailed response. Most of my concerns about the comparison with SF-based and FB have been addressed. The modifications to FB and the results for the biased sampling in FB shall be included as a sanity check.
> >
> > Follow-up questions:
> >
> > **Regarding the explanation on why PSM is a better approach than FB**
> > >(1) FB inherits limitations from SF-based approaches described above due to a similar linear mapping. (2) During training, FB samples from policies that are optimized (through Q function maximization) for the corresponding reward ($B^{-1}x$). This maximization step is prone to overestimation. (3) FB is unstable due to optimizes for a changing reward function as B gets updated.
> >
> > I agree with (1). That said, PSM also ultimately relies on linear mapping (Corollary 4.2), right? Specifically, if the state and action spaces are continuous, then using a set of finite-dimensional basis also presumes an approximation based on linear mapping (Section 5.1). Please correct me if I missed anything.
> >
> > As for (2), has this been observed in practice? Moreover, even if overestimation indeed occurs, that is a consequence of Q-learning, not the FB decomposition itself. There are already many existing techniques that can mitigate the overestimation issue. That’s why this point is not that convincing in my opinion.
> >
> > Regarding (3), if I understand it correctly, PSM can also have the similar issue, i.e., when $\Phi$ and $b$ get updated during training, the resulting $w^{\pi}$ has to change accordingly if the occupancy measure has to be matched by $\Phi w^{\pi}+b$ (Equation (7)). Can the authors comment on this?
> >
> > **Regarding the discrete codebook**
> >
> > Thanks for the response. I believe it would be helpful to describe how to determine the codebook size and how it affects the performance as this appears to be an important hyperparameter in practice.

---

> > > ### Author Response · Authors · 2025-04-07
> > >
> > > We thank the reviewer for their response. We address all the concerns raised by the reviewer  as follows:
> > >
> > > > The modifications to FB and the results for the biased sampling in FB shall be included as a sanity check.
> > >
> > > We performed this experiment, and the errors are: 2.13 +- 0.73 (gridworld) and 12.9 +- 1.63 (four room). While these are better than the adapted FB, they are still slightly worse than PSM: 2.05 +- 1.20 (gridworld) and 11.54 +- 1.07. We shall include these results in the camera-ready.
> > >
> > > > I agree with (1). That said, PSM also ultimately relies on linear mapping (Corollary 4.2), right? Specifically, if the state and action spaces are continuous, then using a set of finite-dimensional basis also presumes an approximation based on linear mapping (Section 5.1). Please correct me if I missed anything.
> > >
> > > We agree with the reviewer that both PSM and FB lose information under finite dimensions. But, the concern that we raise for FB (and all SF-based works) also holds when dimensions are infinite. While PSM projects successor measures to a linear space, the mapping between rewards and policies is not linear, and is obtained after solving a linear program. On the contrary, FB (and SF-based methods) assume a one-to-one linear relationship between policy embeddings and reward functions.
> > >
> > > > As for (2), has this been observed in practice? Moreover, even if overestimation indeed occurs, that is a consequence of Q-learning, not the FB decomposition itself. There are already many existing techniques that can mitigate the overestimation issue. That’s why this point is not that convincing in my opinion.
> > >
> > > PSM not only differs from FB in its decomposition but also in the training. While both these methods use off-policy learning, PSM uses a relatively mild form of off-policy learning that is more stable than the one derived by maximization. Similar observations have been made by [1]. FB relies on the Q-maximization as it uses the actor learnt from this maximization to generate samples for learning successor measures. PSM on the other hand does not use a learnt actor for its off-policy evaluation, hence being more immune to the overestimation issues.
> > >
> > > [1]: Farebrother, Jesse, et al. "Proto-Value Networks: Scaling Representation Learning with Auxiliary Tasks." The Eleventh International Conference on Learning Representations.
> > >
> > > > Regarding (3), if I understand it correctly, PSM can also have the similar issue, i.e., when $\Phi$ and $b$ get updated during training, the resulting $w^{\pi}$ has to change accordingly if the occupancy measure has to be matched by $\Phi w^{\pi}+b$ (Equation (7)). Can the authors comment on this?
> > >
> > > Yes, we agree that the resulting $w^\pi$ has to also change along with $\Phi$ and $b$, but in practice all these networks are trained concurrently. This is the same as training $F$ and $B$ jointly in FB. This step can be thought of as decomposing the representation of $M$ based on the representative biases. Given an input $\pi$ (uniquely by integer code) we learn $\Phi, w$ and $b$ jointly to optimize for a fixed target of successor measure of $\pi$.
> > >
> > > The difference with FB is that FB uses an actor that is trained to optimize reward $B^{-1}z$, to sample actions when learning successor measures for policy $\pi_z$. As $B$ is constantly updated, the reward changes making the actor optimization unstable. This instability is absent in PSM as PSM performs off-policy evaluation using samples from the codebook policy.
> > >
> > > >  It would be helpful to describe how to determine the codebook size and how it affects the performance as this appears to be an important hyperparameter in practice.
> > >
> > > We will be adding an ablation on the codebook size in the camera ready. The codebook size used for are experiments is $2^{16}$ that is, the integer codebook representation $z \leq 2^{16}$. These representations are implemented as binary strings and converted to integers when queried as the seed for sampling.

---

### Decision · Program_Chairs · 2025-05-01

**Decision:**

Accept (poster)

**Comment:**

In this work the authors study the problem of representation learning in reinforcement learning tasks where they develop methods for zero shot learning. Based on the observation that successor measures satisfy a Bellman equation that is independent of the policy they introduce the Proto Successor Measure (PSM). This construction is based on the observation that any solution to an affine equation can be represented as an affine combination of a basis set, and therefore one can represent a successor measure by a corresponding set of linear weights. This work’s conceptual and theoretical contributions are accompanied with experiments in gridworld and control environments showing the benefits of their approach.